# OPEN

# *Pseudomonas putida* mediates bacterial killing, biofilm invasion and biocontrol with a type IVB secretion system

Gabriela Purtschert-Montenegro[1,3], Gerardo Cárcamo-Oyarce[1,2,3], Marta Pinto-Carbó[1], Kirsty Agnoli[1], Aurélien Bailly[1] and Leo Eberl [1]✉

**Many bacteria utilize contact-dependent killing machineries to eliminate rivals in their environmental niches. Here we show that the plant root colonizer *Pseudomonas putida* strain IsoF is able to kill a wide range of soil and plant-associated Gram-negative bacteria with the aid of a type IVB secretion system (T4BSS) that delivers a toxic effector into bacterial competitors in a contact-dependent manner. This extends the range of targets of T4BSSs—so far thought to transfer effectors only into eukaryotic cells—to prokaryotes. Bioinformatic and genetic analyses showed that this killing machine is entirely encoded by the *kib* gene cluster located within a rare genomic island, which was recently acquired by horizontal gene transfer. *P. putida* IsoF utilizes this secretion system not only as a defensive weapon to kill bacterial competitors but also as an offensive weapon to invade existing biofilms, allowing the strain to persist in its natural environment. Furthermore, we show that strain IsoF can protect tomato plants against the phytopathogen *Ralstonia solanacearum* in a T4BSS-dependent manner, suggesting that IsoF can be exploited for pest control and sustainable agriculture.**

Many reports have demonstrated the great potential of microbial inoculants for bioremediation, biofertilization and biocontrol applications[1–4]. However, the lab-to-field transition remains a major limiting factor since good in vitro performance is rarely reproduced in field trials. Microbial survival, establishment and colonization are key features for biocontrol agents and applied microorganisms are often unable to persist in the environment or are rapidly outcompeted[5–8]. One important reason for the failure of a strain to colonize a desired niche is that the planktonic inoculants are unable to invade and persist in indigenous microbial consortia, which live within biofilms[5,9,10]. Biofilm cells are embedded in an extracellular matrix that protects them from external stresses, such as nutrient limitation, predation and the host immune response. The extracellular matrix also restricts the entry of invaders into the biofilm interior, and while bacteria can colonize and grow on the biofilm exterior, they are readily removed by shear forces[11–14]. Additionally, members of the indigenous biofilm consortium have evolved an arsenal of defence strategies that limit the successful establishment of inoculants in the rhizosphere[15–17]. Bacteria display two main strategies to antagonize invaders: the release of small molecules with antimicrobial activity into their surroundings and the delivery of toxic effector proteins through diverse secretion systems into neighbouring opponents, which relies on cell-to-cell contact[18–21]. The most recent type is homologous to type IVA secretion systems (T4ASS) in *Xanthomonas citri* and *Stenotrophomonas maltophilia*, which deliver toxic effectors that cause lysis of susceptible competitor cells[22–24]. There are two main classes of T4SS: (1) type IVA secretion systems (T4ASS), mostly used for DNA delivery and exemplified by the VirB/D4 system of *Agrobacterium tumefaciens* and (2) type IVB secretion systems (T4BSS), mostly used to deliver effector proteins into eukaryotic hosts as exemplified by the Dot/Icm system found in intracellular pathogens such as *Legionella pneumophila*[25,26]. The two classes are only distantly related and T4BSS assemblies are larger than T4ASSs, comprising 27 components for the *L. pneumophilla* Dot/Icm system compared with 12 components for the VirB/D4 system[26].

In this study, we identified a T4BSS that can deliver toxic effectors into bacterial competitors, breaking the paradigm that T4BSSs are only used for effector transfer into eukaryotic cells[25–27]. This bacterial killing machine is encoded by a rare genomic island in the plant beneficial bacterium *Pseudomonas putida* IsoF[28,29] and allows the strain not only to invade existing biofilms but also to protect tomato plants against the pathogen *R. solanacearum*.

## Results

**IsoF kills a wide range of Gram-negative bacteria.** When *P. putida* IsoF (marked with green fluorescent protein (Gfp)) and *P. putida* KT2442 (marked with mCherry) were co-inoculated in a 1:1 ratio on a minimal medium plate, no red fluorescence could be observed from the microcolony after 24 h, indicating that KT2442 had been killed (Fig. 1a). We determined the colony forming units (c.f.u.s) of the two strains and found that after 2 d, IsoF had completely eliminated KT2442 (Fig. 1a). No adverse effect was seen when the two strains were separated by a 0.2 µm pore size filter, suggesting that killing depends on cell-to-cell contact (Extended Data Fig. 1). To obtain further insight into the underlying molecular mechanism, we performed contact-dependent killing (CDK) experiments on plates supplemented with propidium iodide (PI) to visualize dead cells. After 24 h, PI staining was observed where the two inoculated cultures overlapped, whereas dead cells were absent from the pure culture regions (Fig. 1b). Time-lapse confocal laser scanning microscopy (CLSM) was used to demonstrate

[1]Department of Plant and Microbial Biology, University of Zurich, Zurich, Switzerland. [2]Present address: Department of Biological Engineering, Massachusetts Institute of Technology, Cambridge, MA, USA. [3]These authors contributed equally: Gabriela Purtschert-Montenegro, Gerardo Cárcamo-Oyarce. ✉e-mail: leberl@botinst.uzh.ch

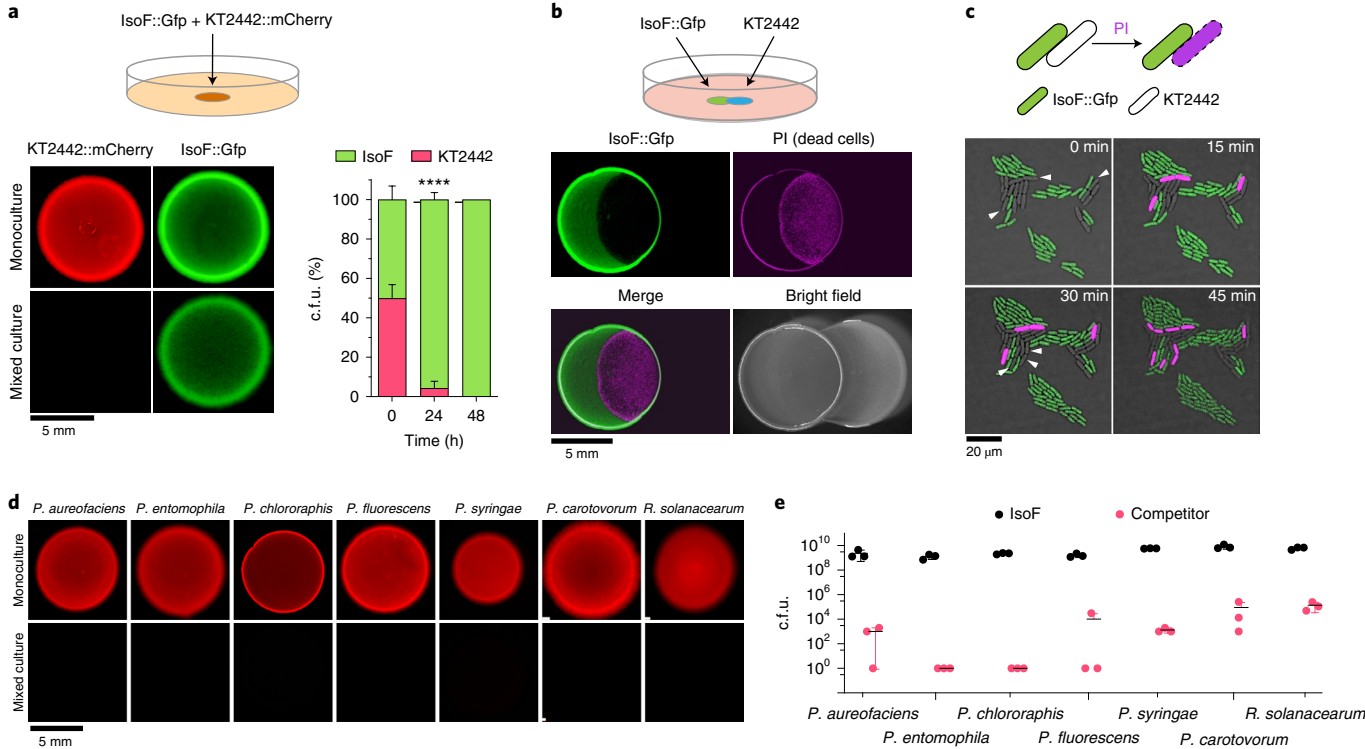

**Fig. 1 | IsoF displays contact-dependent antagonism against a wide range of Gram-negative bacteria. a**, Left: IsoF::Gfp kills KT2442::mCherry after co-inoculation on ABC agar plates. Fluorescence is indicative of viable bacteria. Right: the percentage of the c.f.u.s of each bacterial population in the mixed culture at 0 h, 24 h and 48 h. Data are mean ± s.d. from 3 biological replicates ($n = 3$). Unpaired $t$-test, ****$P < 0.0001$. **b**, IsoF antagonism is restricted to areas where IsoF::Gfp and KT2442 colonies are in direct contact. The medium was supplemented with PI to visualize dead cells. **c**, IsoF::Gfp kills KT2442 cells in a contact-dependent manner. Cell death was monitored by PI staining (cells shown in magenta). **d**, IsoF kills a wide range of Gram-negative plant-associated bacteria, including *P. aureofaciens*, *P. entomophila*, *P. chlororaphis*, *P. fluorescens*, *P. syringae*, *P. carotovorum* and *R. solanacearum*. All competitors were tagged with mCherry. **e**, C.f.u.s were determined after 24 h of co-inoculation. Data are mean ± s.d. from 3 independent biological replicates ($n = 3$). Representative fluorescence images of 3 independent experiments are shown.

that KT2442 cells were killed after they had been in direct contact with IsoF::Gfp (Fig. 1c and Supplementary Video 1). We noticed that dead cells did not lyse or change their morphology. To determine the range of target species antagonized by IsoF, we fluorescently marked several soil- and plant-associated bacteria as well as some phytopathogens, and tested their susceptibility to killing by IsoF. All tested strains were killed after 24 h of co-culture with IsoF (Fig. 1d,e), suggesting that IsoF possesses a highly efficient, broad host-range, CDK machinery.

**IsoF utilizes a rare T4BSS for bacterial killing.** To identify the mechanism responsible for contact-dependent killing by IsoF, we constructed a mini-Tn5 transposon insertion library of this strain and tested about 5,000 mutants for their ability to kill *P. aureofaciens*::mCherry, a strain that is easy to cultivate and is highly susceptible to IsoF-mediated killing, when grown as mixed macro-colonies (Fig. 2a). The analysis of eight mini-Tn5 insertion mutants defective in killing *P. aureofaciens*::mCherry identified four genes within a large gene cluster (designated *kib* for killing, invasion, biocontrol, see below) that encodes several elements of a T4BSS (Fig. 2a,b and Extended Data Fig. 2a). While T4BSSs of intracellular pathogens are known for their capacity to deliver effectors to their eukaryotic hosts[26,30], they have so far not been reported to be involved in interbacterial killing. To validate the results of the mutant screen, we constructed defined T4BSS mutants: Δ*dotHGF*, which lacks the main structural components of the secretion channel, was unable to kill *P. aureofaciens* or KT2442. Likewise, inactivating the region spanning *PisoF_02323* to *PisoF_02325*, which

encodes hypothetical proteins, TrbN and DotD, prevented IsoF from killing other bacteria (Fig. 2c,d and Extended Data Fig. 2b).

The web-based tool ICEfinder[31] identified the *kib* cluster as part of a genomic island and defined its borders with genes *PisoF_02313* and *intA_3*, in agreement with the lower guanine-cytosine (GC) content of the *kib* locus (58.8%) compared with the average GC content of the IsoF genome (62.6 %) (Supplementary Table 3). This island has a size of 66,917 bp and encodes 61 genes, 17 of which share homology with described T4BSS structural genes, 37 were defined as hypothetical proteins, and 4 encode a type I restriction modification (RM) system and an integrase at the 3'-end (Fig. 2b,c and Supplementary Table 4). A BLAST search revealed that *kib* genes are present in 11 *Pseudomonas* strains, 8 of which were classified as *P. putida* (Supplementary Table 3). Interestingly, all orthologous *kib* gene clusters showed conserved synteny and were located at the same chromosomal position, suggesting a common ancestor (Extended Data Figs. 3 and 4). Notably, the orthologous clusters showed deletions at the 3'-end of the island, including the RM system and the flanking integrase. In conclusion, while the *kib* island is found in few *Pseudomonas* strains, it is highly conserved and appears to encode all components of the bacterial killing machine, suggesting that it was recently acquired via horizontal gene transfer.

**The *kib* gene cluster encodes an effector-immunity (E-I) pair.** Contact-dependent killing systems deliver toxic effector molecules into bacterial competitors. To avoid self-killing, the attacking bacterium produces a cognate immunity protein that neutralizes the toxin. The immunity and toxin genes form a so-called E-I pair and

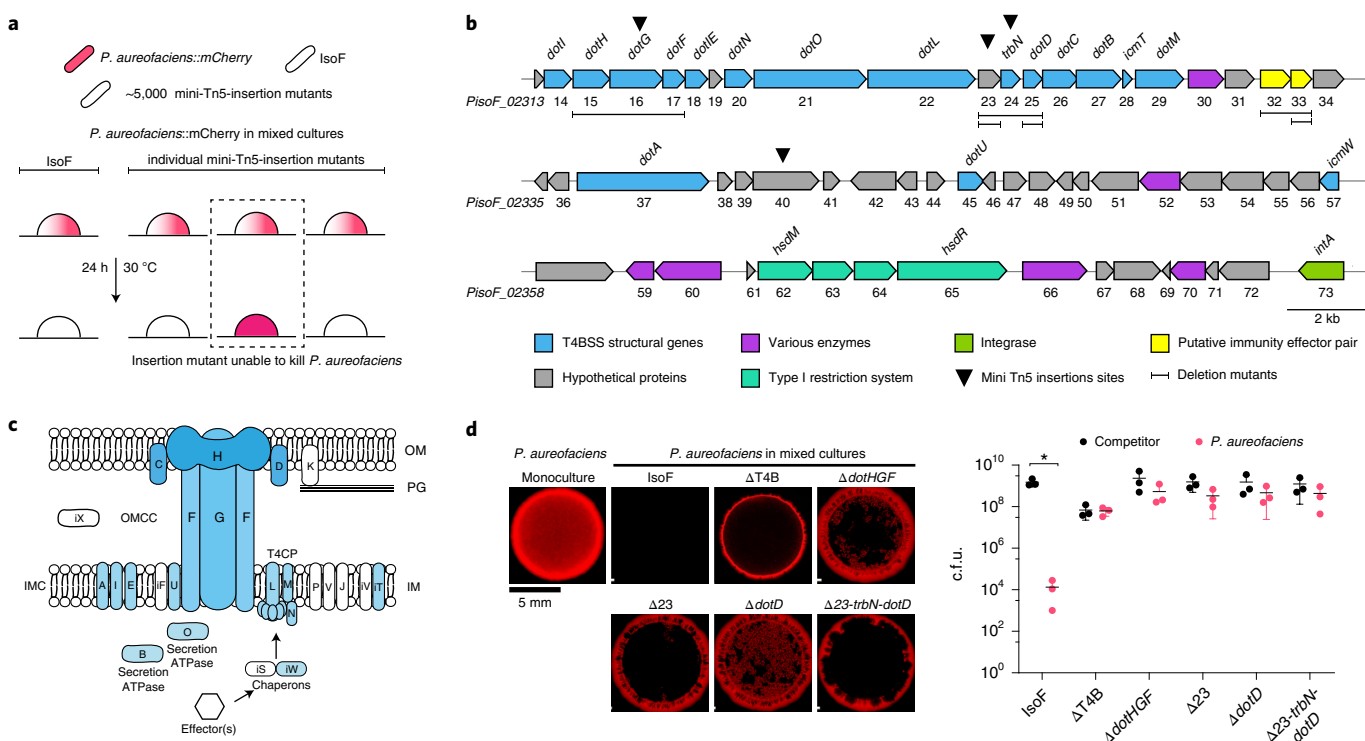

**Fig. 2 | The IsoF T4BSS-dependent killing machine is encoded on a GI. a**, Single IsoF-Tn5-insertion mutants were mixed with *P. aureofaciens* tagged with mCherry. Red fluorescence indicates the loss of killing activity by the mutant. Dashed rectangle represents an insertion mutant defective in killing. **b**, Genetic organization of the IsoF GI that encodes a T4BSS required for bacterial killing. The cluster has a length of 69.9 kb and codes for 17 T4BSS structural and 34 hypothetical proteins. Additionally, the 3′ end encodes a type I RM system and an integrase. Regions that were deleted in defined mutants are underlined in black: Δ*dotHGF*, Δ23, Δ*dotD*, Δ23-*trbN-dotD*, Δ33 and Δ32-33. **c**, Architecture of the T4BSS gene cluster in *Legionella*. Homologues of proteins highlighted in blue are encoded by the IsoF GI. In the *Legionella* Icm/Dot system, effector molecules bind to chaperons that interact with the type 4 coupling protein (T4CP) before the effector molecules are translocated through the outer membrane core complex. The two ATPases provide energy and interact with the inner membrane complex (IMC) at the substrate recognition and translocation domains (modified from refs. [25,74]). **d**, Left: fluorescence images showing *P. aureofaciens*::mCherry in competition with the IsoF wild type and various deletion mutants after 24 h of co-incubation. Right: c.f.u.s of the two competing strains. Data are mean ± s.d. of 3 independent biological replicates (*n* = 3). Unpaired *t*-test, \**P* < 0.05.

are often co-transcribed[18,22–24,32]. To investigate whether an E-I pair was present within the *kib* region, we deleted 49.5 kb of the genomic island containing all *kib* genes (*PisoF_02313* to *PisoF_02360*, Extended Data Fig. 4). The resulting mutant, ΔT4B, no longer killed *P. aureofaciens* or KT2442 and formed mixed macrocolonies with both strains (Fig. 2d and Extended Data Fig. 2b). We hypothesized that the absence of an E-I pair would render the ΔT4B mutant sensitive to the wild-type strain, while its presence would confer resistance. CDK assays revealed that the mutant strain was killed, while it co-existed with the Δ*dotHGF* mutant (Fig. 3a), demonstrating that the genes required for killing and self-protection are present within the *kib* cluster. Moreover, IsoF was unable to kill mutants Δ*dotHGF* and Δ23-*trbN-dotD*, indicating that genes conferring immunity are not located within the deleted regions (Extended Data Fig. 5).

**Identification of an immunity gene by transposon sequencing.**
Immunity genes are essential since cells lacking an immunity protein would either be killed by neighbouring bacteria or die due to self-intoxication[32,33]. Hence, it is possible to identify potential E-I pairs by transposon sequencing (Tn-seq)[33,34]. We generated a saturated transposon insertion library in IsoF and subjected the library to three different growth regimes: (1) growth in liquid medium with shaking to prevent cell-to-cell contact, (2) growth on an agar surface either alone or (3) in the presence of the competitor *P. aureofaciens* to promote killing (Fig. 3b and Supplementary Table 5). Our analysis revealed that *PisoF_02332* was virtually devoid of transposon

insertions in all three treatments (Fig. 3c and Supplementary Table 6). This gene appears to be co-transcribed with *PisoF_02333*, possibly constituting an E-I pair. In silico analysis of PisoF_02332 and PisoF_02333 predicted the subcellular location of both proteins to be the cytosol. In agreement with this prediction, but in contrast to the finding that the X-T4ASS immunity proteins of *X. citri* and *S. maltophilia* are localized in the periplasm[22,23], we were not able to identify either a signal peptide sequence at the N terminus of PisoF_02332 or translocation signals of known *Legionella* effectors in the C-terminal region of the protein[22,23,26,35] (Extended Data Fig. 6a). Interestingly, we found that PisoF_02332 has a C-terminal FxxxLxxxK motif, which is a known recognition sequence of the *Legionella* T4BSS[35], suggesting that the immunity protein may be transferred together with its cognate effector. Comparison of phylogenetic trees constructed using (1) the *PisoF_02332-33* genes, (2) all orthologues of the *kib* cluster and (3) eight housekeeping genes from the strains carrying the *kib* locus, revealed congruent tree topology, suggesting that these strains form a defined lineage that originated from a common ancestor (Extended Data Fig. 7).

To assess the role of this putative E-I pair in bacterial killing, we deleted both genes (*PisoF_02332* and *PisoF_02333*) in IsoF to generate Δ32-33. We were also able to delete the putative effector gene alone, giving rise to mutant Δ33. We noticed that Δ32-33 grew slower on ABC minimal media relative to the parental strain or mutant Δ33 (Supplementary Fig. 1). To establish a fair competition situation despite the growth difference, the CDK assays with

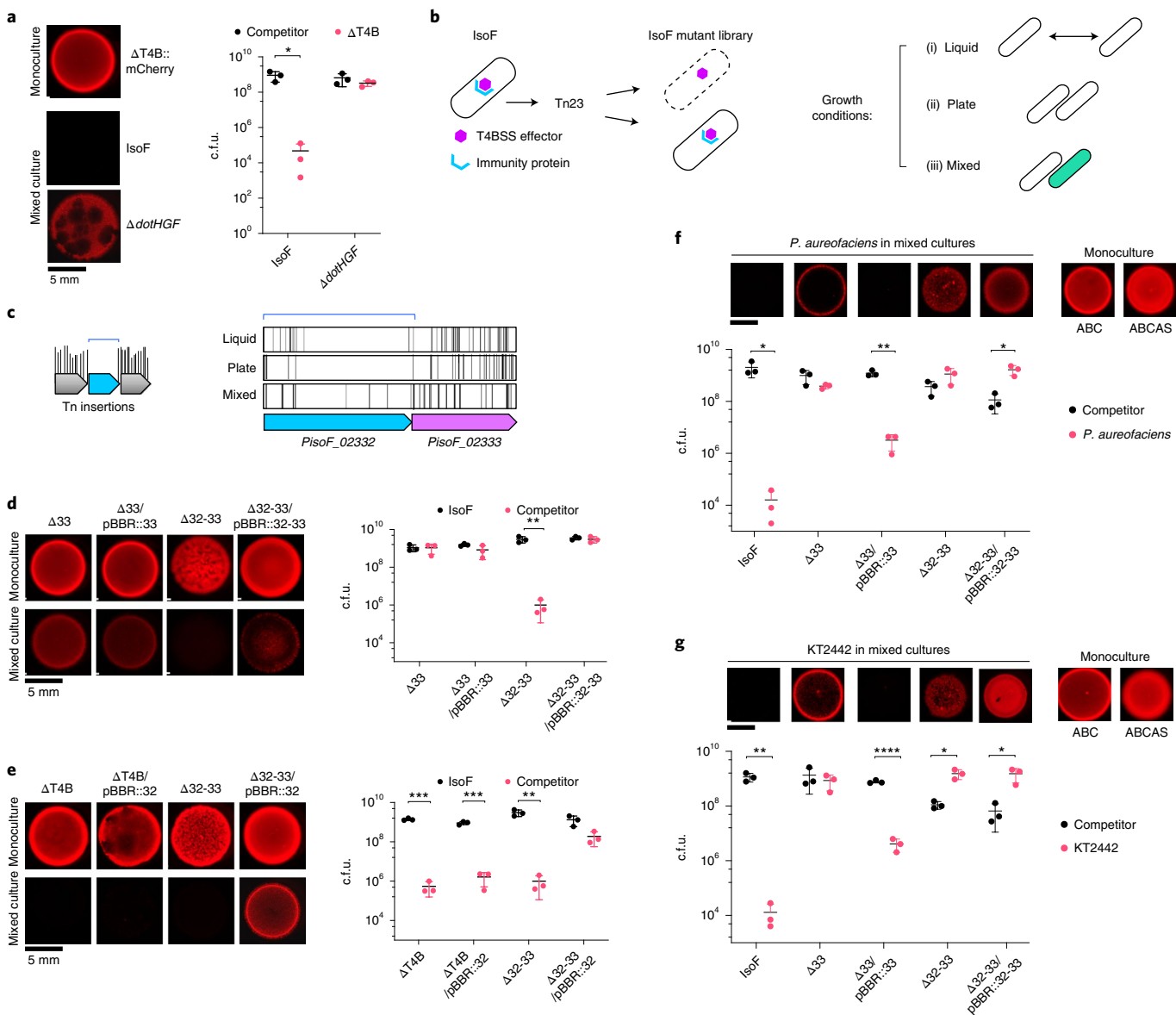

**Fig. 3 | An E-I pair is encoded within the *kib* gene cluster. a**, Left: contact-dependent killing of the ΔT4B mutant against the IsoF wild type and the Δ*dotHGF* deletion mutant. Right: c.f.u.s of the competing strains determined after 24 h of co-inoculation. **b**, An effector will be toxic for the cell in the absence of its cognate immunity protein. The following conditions were used to challenge an IsoF Tn*23* mutant library: (i) growth in liquid medium with shaking to prevent cell-to-cell contact, (ii) growth on an agar surface either alone or (iii) in the presence of the rival strain *P. aureofaciens* to promote competition (mixed). **c**, The unique insertion density approach of the Tn-Seq Explorer software was used to identify genes that provide a fitness benefit for growth under different growth conditions[65]. Left: *PisoF_02332* (blue) was found to have very few transposon insertions under all growth conditions tested. Right: a putative effector gene, *PisoF_02333* (magenta), is located downstream of *PisoF_02332*. The Tn*23* insertions in the two genes for the three growth conditions are shown. **d**, Left: CDK of IsoF against the deletion mutants Δ32-33 (lacking the E-I pair), Δ33 (lacking the effector gene) and their complemented derivatives. Right: c.f.u.s of the competing strains determined after 24 h of co-inoculation. **e**, CDK of the IsoF wild type against mutants ΔT4B/pBBR::32 and Δ32-33/pBBR::32. Right: c.f.u.s of the competing strains determined after 24 h of co-inoculation. **f,g**, CDK of KT2442 and *P. aureofaciens*, respectively, against the deletion mutants Δ32-33 (lacking the E-I pair), Δ33 (lacking the effector gene) and their complemented derivatives. Top: representative fluorescence images of three independent experiments are shown. Scale bars, 5 mm. Bottom: c.f.u.s of the competing strains were determined after 24 h incubation. Data are mean ± s.d. of 3 independent replicates (*n* = 3). Unpaired *t*-test, **P* < 0.05; ***P* < 0.01; ****P* < 0.001; *****P* < 0.0001.

Δ32-33 were performed on ABC medium supplemented with casamino acids and the c.f.u.s were normalized to the number of cells recovered from the monoculture of Δ32-33 after 24 h. Importantly, Δ33 was unable to compete with *P. aureofaciens* or KT2442 and complementation partially rescued the killing phenotype, suggesting that *PisoF_02333* encodes a toxic effector protein (Fig. 3f,g). This is further supported by the lack of killing of *P. aureofaciens* and KT2442 by the double mutant Δ32-33, although we were

unable to restore killing by complementation. We used sodium dodecyl-sulfate polyacrylamide gel electrophoresis (SDS–PAGE) to investigate expression in the complemented strain. While we observed a band corresponding to the immunity protein, we could not detect the toxin, indicating that the immunity protein is produced in excess compared with the toxin (Extended Data Fig. 6b). We therefore hypothesize that killing was not restored because of an unphysiological overexpression of the immunity protein in the

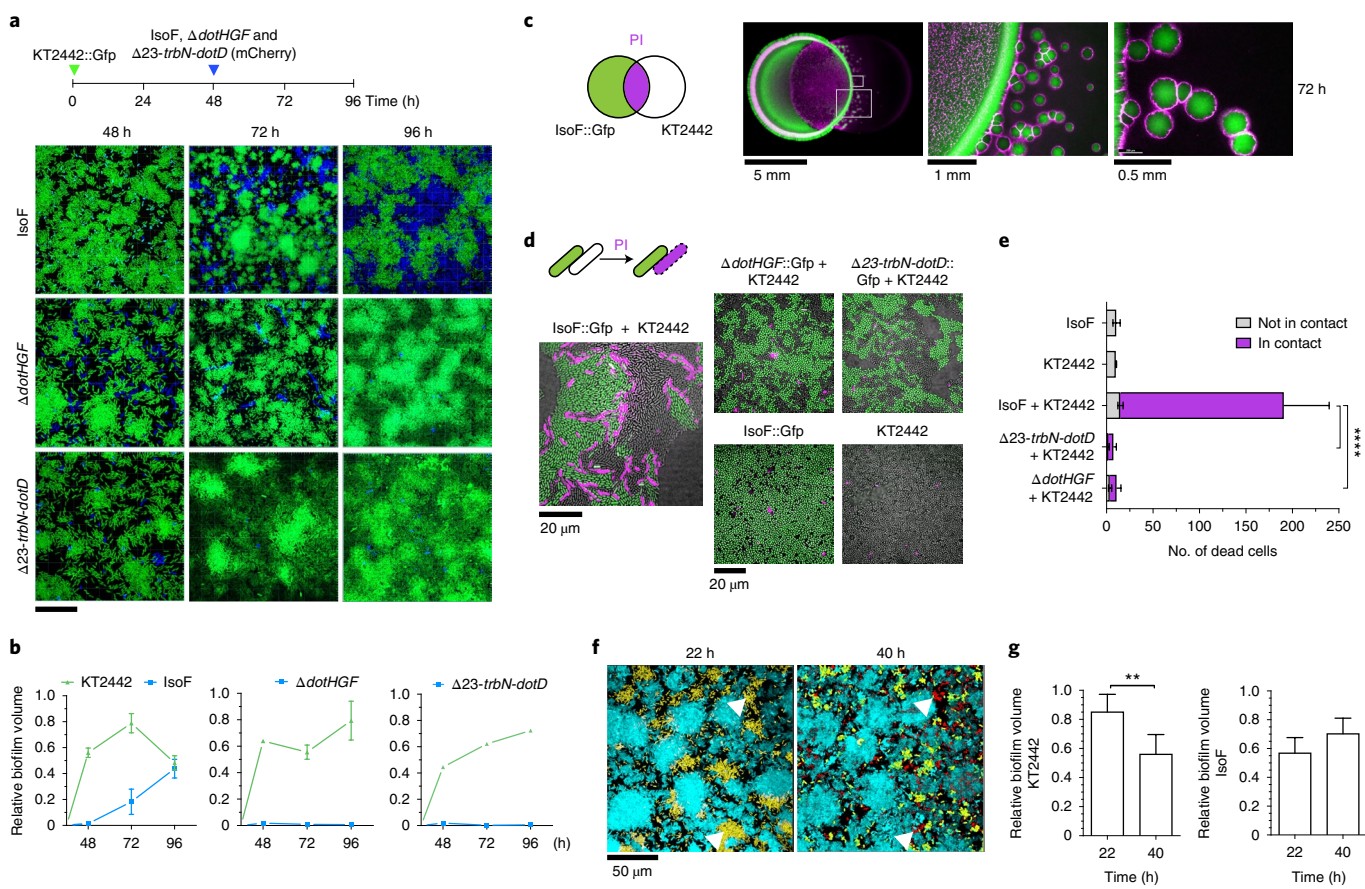

**Fig. 4 | IsoF wild type invades and displaces a pre-established KT2442 biofilm. a**, A 2-day-old biofilm of KT2442::Gfp (green) is invaded by IsoF::mCherry (blue) but not by Δ*dotHGF*::mCherry (blue) or Δ23-*trbN-dotD*::mCherry (blue). Scale bar, 30 μm. **b**, Relative biofilm biomass (volume) of KT2442::Gfp relative to IsoF::mCherry, Δ*dotHGF*::mCherry and Δ23-*trbN-dotD*::mCherry. Data are mean ± s.d. from up to 3 biological replicates (*n* = 3). **c**, Macrocolony of KT2442 is gradually colonized by IsoF::Gfp expanding microcolonies. The media were supplemented with PI to visualize dead cells (magenta). **d**, CDK of IsoF, Δ*dotHGF* and Δ23-*trbN-dotD* (green) against KT2442 after 18–20 h of incubation. Monocultures of IsoF::Gfp and KT2442 are indicated. Dead cells were visualized by staining with PI. **e**, The number of dead cells in contact and not in contact with a green fluorescent cell were quantified from at least 9 randomly chosen images from the CDK experiments. As a control, the strains were also inoculated without a competitor and the number of dead cells was determined. Data are mean ± s.d. of 3 independent replicates (*n* = 3). Unpaired *t*-test, **** *P* < 0.0001. **f**, Mixed biofilm of KT2442::Gfp (yellow) and IsoF::Cfp (cyan) after 40 h of co-cultivation. White arrowheads indicate the same positions at 22 h and 40 h, where dead cells (red) were visualized by staining with PI. **g**, Relative biofilm volumes of KT2442 and IsoF at 22 and 40 h. Unpaired *t*-test, ***P* < 0.01 (*n* = 3).

complemented strain that effectively neutralized the effector. In competition against the IsoF wild type, mutant Δ33 and its complemented derivative survived, indicating that both strains are immune to the IsoF effector toxin. By contrast, mutant Δ32-33 was killed by IsoF, while the complemented strain co-existed with IsoF, implying that *PisoF_02332* confers immunity to *kib*-mediated killing (Fig. 3d). To further test this, mutants ΔT4B and Δ32-33 were complemented with *PisoF_02332* on a plasmid (pBBR::32) and the resulting strains were used in killing assays against IsoF. While strain Δ32-33/pBBR::32 co-existed with IsoF, mutant ΔT4B/pBBR::32 was killed (Fig. 3e). SDS–PAGE analysis showed that PisoF_02332 is not expressed in the ΔT4B mutant background, explaining its sensitivity to *kib*-mediated killing and suggesting that the *kib* gene cluster encodes functions required for the expression of PisoF_02332 or affects its stability (Extended Data Fig. 6b).

**The *kib* system enables IsoF to invade an established biofilm.** In our CDK assays, growth of IsoF was restricted to the initial inoculation area after 24 h of incubation, probably because dead cells created a barrier that prevented further killing[36] (Fig. 1b). However, upon prolonged incubation of CDK plates, we observed that IsoF began to invade the space occupied by the target strain and formed

satellite colonies after 72 h (Fig. 4c and Supplementary Fig. 5). We hypothesized that killed cells eventually lysed and no longer constituted a barrier against invasion, indicating that contact-dependent killing might be an efficient way to eliminate competitors in polymicrobial biofilms. To evaluate the role of *kib*-mediated killing in mixed-species biofilms, we decided to use IsoF and KT2442, which are both good biofilm producers. To this end, we first established a KT2442::Gfp (green) biofilm in a flow-cell system and then introduced IsoF::mCherry (blue) (Fig. 4a). Within 1 d, IsoF cells attached to the surface began to proliferate and formed numerous microcolonies. After 2 d, IsoF had formed a mature biofilm by invading and displacing the KT2442 biofilm. The volume of KT2442 biofilm decreased by approximately 40% between 24 and 48 h post IsoF inoculation, which reached equal biomass with KT2442 after 2 d of competition (Fig. 4b). Without competition, the biomass of the KT2442 biofilm increased steadily over time (Extended Data Fig. 8a,c). When a pre-established KT2442 biofilm was challenged with the *kib* mutants Δ*dotHGF* or Δ23-*trbN-dotD*, neither mutant was able to form microcolonies or invade the existing biofilm (Fig. 4a,b). Importantly, Δ*dotHGF* and Δ23-*trbN-dotD* mutants in isolation formed biofilms similar to the IsoF wild-type strain (Extended Data Fig. 8b,d). We hypothesized that IsoF employed

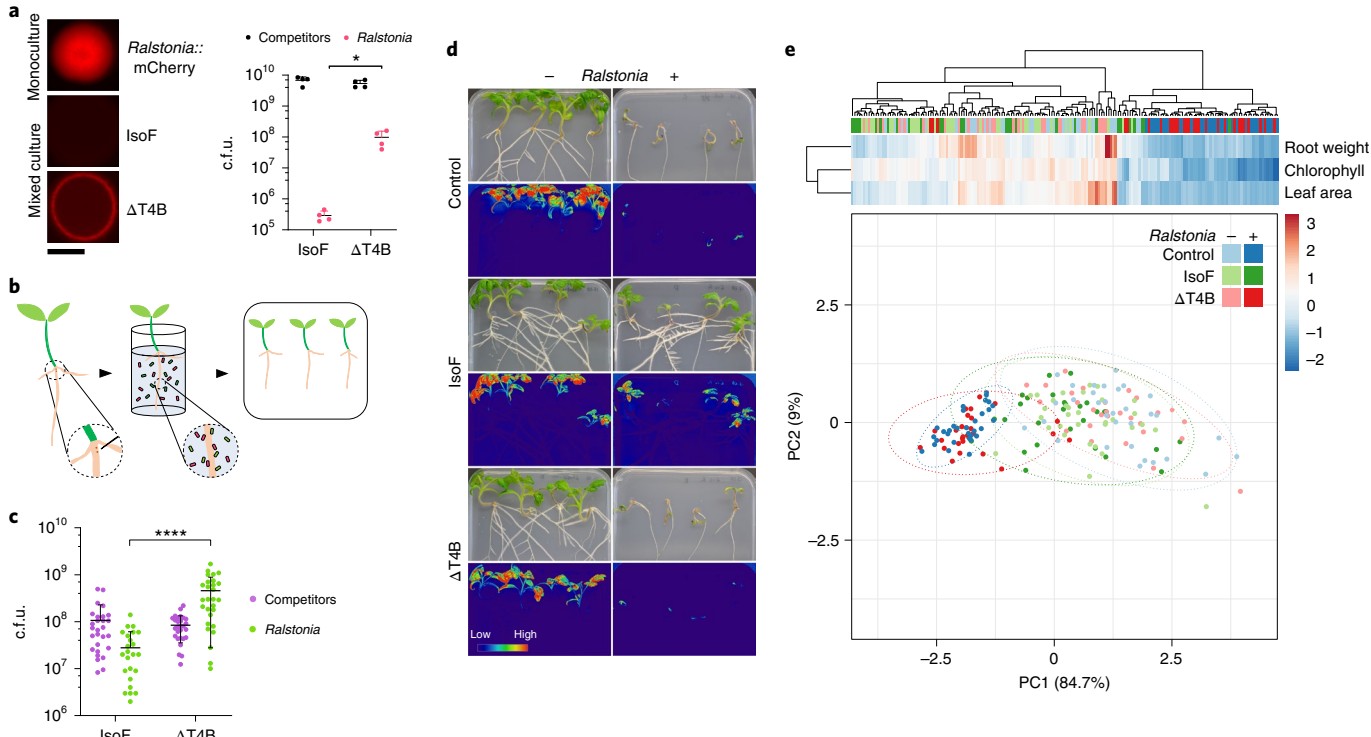

**Fig. 5 | IsoF T4BSS protects tomato plants against bacterial wilt in vitro. a**, Left: CDK of IsoF and ΔT4B against *R. solanacearum* tagged with mCherry. Right: c.f.u.s of the competing strains determined after 24 h of co-inoculation. Data are mean ± s.d. from 4 biological replicates (*n* = 4). Unpaired *t*-test, *\**P < 0.05. Scale bar, 5 mm. **b**, Schematic representation of the experimental approach used. Seedlings were injured twice with a needle at the root-shoot junction and immersed in the bacterial suspensions. Rod-shaped dots indicate green and pink bacteria. Inoculated seedlings were transferred to ½MS plates for further growth. **c**, C.f.u.s of recovered bacterial cells from the tomato roots. In total, 28 plants were assessed, with a minimum of 8 plants per treatment for each of the 3 independent replicates (*n* = 3). C.f.u.s of monocultures are in Supplementary Fig. 6. Data are mean ± s.d. Unpaired *t*-test, *\*\*\*\**P < 0.0001. **d**, Representative images of in vitro tomato plants 22 d after inoculation. False-colour pictures display chlorophyll contents estimated from RGB values. **e**, Principal component analysis (bottom) and hierarchical clustering heat map (top) of estimated plant health parameters. Colour scale (dark orange to blue) indicates the *Z*-Score values of the measured parameters (root weight, chlorophyl content, leaf area).

its T4BSS to kill KT2442 cells upon contact, creating space for the expansion of the IsoF biofilm. To test this, we inoculated flow cells with IsoF::Cfp (cyan) and KT2442::Gfp (yellow) with equivalent numbers of cells and monitored the fate of KT2442 microcolonies neighbouring IsoF microcolonies by adding PI (red) as an indicator of cell death. Dead cells were observed at positions where the two strains were in direct contact (Fig. 4d,f). We determined that the biofilm volume of KT2442 was reduced by approximately 20% (Fig. 4g). We next visualized killing of KT2442 by IsoF::Gfp in a mixed monolayer biofilm on the surface of a minimal medium agar pad (Fig. 4d). After 18 h, we observed that 92% of the dead KT2442 cells (magenta) were located next to IsoF::Gfp (green) cells, demonstrating that *kib*-mediated killing is strictly dependent on cell-to-cell contact (Fig. 4e). When KT2442 was challenged with either the Δ*dotHGF* or the Δ23-*trbN-dotD* mutant (green), very few dead cells were observed, similar to monoculture biofilm controls (Fig. 4d,e). These results suggest that the *kib* system not only allows IsoF to defend itself against competitors but also to kill bacteria that live within an established biofilm community.

**The *kib* locus is required for biocontrol activity of IsoF.** In vitro competition experiments showed that IsoF killed *R. solanacearum*, while the ΔT4B mutant did not (Fig. 5a). To assess whether the *kib* killing system could be useful for biocontrol applications, we tested whether IsoF is able to protect tomato plants from infection with *R. solanacearum*, a major pathogen causing bacterial wilt in a wide range of crops[37]. Considering that *R. solanacearum* is a soil-borne

pathogen that enters the plant through natural openings such as emerging lateral roots or wounds[38], we injured established tomato seedlings with small incisions (Fig. 5b). At 22 d post infection, control plants inoculated with *R. solanacearum* were severely wilted, with signs of chlorosis and arrested development of the root and shoot systems. By contrast, 90% of the seedlings inoculated with a mixture of *R. solanacearum* and IsoF showed no signs of wilting (Fig. 5d). However, when seedlings were co-inoculated with a mixture of *R. solanacearum* and the *kib* mutant ΔT4B, wilting and underdevelopment were observed in 85% of the plants, indicating that IsoF prevented *R. solanacearum* from spreading into the plant tissues by *kib*-mediated killing. To precisely evaluate wilt development, we determined the chlorophyll content and measured shoot area and root weight of individuals from the treatment groups as a proxy for plant health (Extended Data Fig. 9). These data were subjected to principal component analysis and hierarchical clustering (Fig. 5e). The first two components accounted for 93.7% of the variance, and a score scatterplot clearly clustered the single inoculation with IsoF and ΔT4B groups together with untreated plants, confirming that the strains do not harm tomato plantlets. Plants co-inoculated with *R. solanacearum* and IsoF preferentially clustered with healthy plants, while those co-inoculated with ΔT4B grouped with *R. solanacearum*-infected plants. This indicates that IsoF decreased the pathogen load in the injured tomato tissues in a *kib*-dependent manner. To verify that IsoF indeed killed *R. solanacearum*, we recovered the bacteria attached to the roots and determined the c.f.u.s. This showed that the number of *R. solanacearum* cells

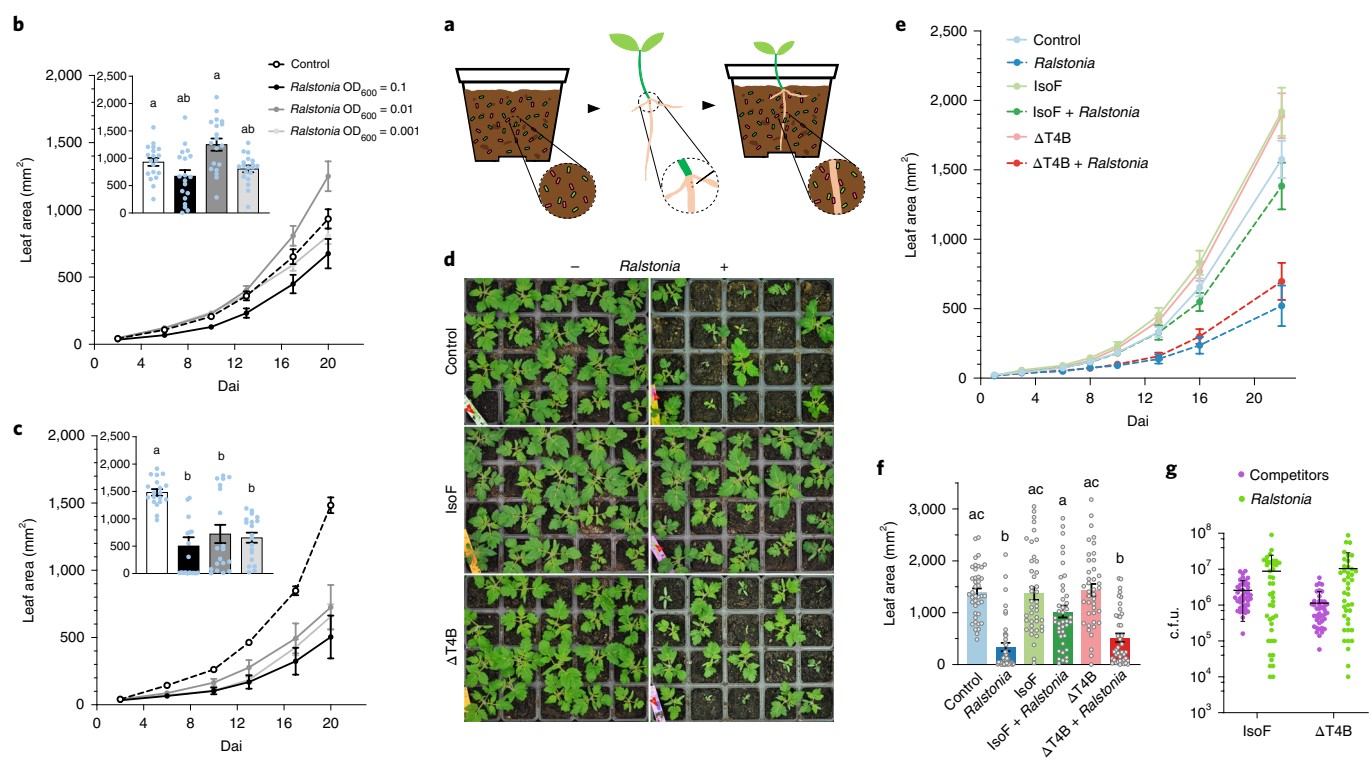

**Fig. 6 | IsoF protects tomato plants grown in non-sterile soil from *R. solanacearum* infection. a**, Schematic representation of the experimental approach used. In this infection model, non-sterile soil is drenched simultaneously with a mixture of IsoF or ΔT4B and *R. solanacearum*. Rod-shaped dots indicate green and pink bacteria. Seedlings were injured twice with a needle at the root-shoot junction and then potted into the drenched soil. **b,c**, Trial assay for uninjured (**b**) and injured (**c**) Micro-Tom seedlings (*n* = 1, 20 seedlings per treatment). Different amounts of Gfp-tagged *R. solanacearum* cells were used to drench the soil, and growth of the plantlets was followed for 20 d. Bar graphs indicate the leaf area at 20 d post inoculation. **d**, Representative pictures from 1 of 3 biological replicates of Micro-Tom plantlets. Pictures were taken 22 d post inoculation. **e**, Leaf area of the plantlets grown on non-sterile soil drenched with a bacterial culture of the different treatments. The growth curves represent the development of the tomato plants of one of the biological replicates. **f**, Bar graph shows the leaf area of the plants 20 d after inoculation (*n* = 3). **g**, C.f.u.s of recovered bacterial cells from the tomato rhizosphere. C.f.u.s of monocultures are in Supplementary Fig. 6. In total, 41 plants were assessed, with a minimum of 10 plants per treatment for each of the 3 independent replicates (*n* = 3), data are mean ± s.d. For each treatment, means with the same letter are not significantly different. ANOVA, Tukey's test, *P* < 0.05.

present after co-inoculation with IsoF was significantly lower than after inoculation with the ΔT4B mutant (Fig. 5c). To evaluate the biocontrol potential of IsoF under more natural conditions, we used a soil-based infection model as detailed in Methods (Fig. 6a). First, we tested the activity of *R. solanacearum* only on non-sterile soil drenched with different c.f.u. amounts employing uninjured and injured seedlings. The dosage of $10^8$ c.f.u. ml$^{-1}$ of *R. solanacearum* on injured seedlings showed the strongest effect on the plantlets where ~66% of the plants did not progress beyond the 2-leaf stage (Fig. 6b,c). When non-sterile soil was drenched with a mixture of IsoF and *R. solanacearum*, 84% of the plants developed as well as the control plants, whereas in the plants co-inoculated with *R. solanacearum* and the *kib* mutant ΔT4B, only 53% of the plants developed their second pair of leaves (Fig. 6d–f). C.f.u. counts recovered from the rhizosphere showed no decrease of *R. solanacearum* c.f.u.s when co-inoculated with either IsoF or the ΔT4B mutant, suggesting that *R. solanacearum* killing might locally prevent plant tissue invasion from the pathogen entry points (Fig. 6g). Collectively, these results demonstrate that the biocontrol capacity of IsoF against *R. solanacearum* depends on the *kib* locus.

## Discussion
While various pathogens use T4BSSs to translocate effector molecules into their eukaryotic host cells[25,26,30], we show here that *P. putida* IsoF uses a T4BSS to kill a wide range of soil and plant-associated Gram-negative bacteria in a contact-dependent manner, extending

the range of organisms targeted by T4BSSs to prokaryotes. Our data suggest that the *kib* locus is part of a recently acquired genomic island that encodes all components of the killing machinery. The *kib* cluster may provide IsoF with a competitive advantage for survival in the environment, as IsoF was shown to be an excellent colonizer of tomato roots[29,39]. IsoF killed various environmental strains, most notably *P. putida* KT2442, which was demonstrated to use its K1-T6SS as an antibacterial killing device[40]. IsoF does not harbour a homologue of the K1-T6SS gene cluster and thus was expected to be sensitive to killing by KT2442. However, when the two wild-type strains were competed against each other, IsoF eliminated KT2442, indicating that T4BSS-mediated killing may occur before KT2442 can fire its T6SS apparatus (Figs. 1a and 3g). Previous work has shown that bacteria have different strategies for deploying their T6SS. While some strains of *Vibrio cholerae* use their T6SS in an untargeted fashion and assemble and fire their apparatus in random locations within the cell, *P. aeruginosa* assembles and fires its organelle only after detecting an attack from another nearby bacterium[41,42], a strategy that has been termed the T6SS tit-for-tat response[43]. More recent work has shown that *P. aeruginosa* senses outer membrane perturbations caused by the attack of competitors, treatment with the membrane-targeting antibiotic polymyxin, or interference with outer membrane biogenesis via a signal transduction pathway that triggers the tit-for-tat response[44]. At present, neither the triggers of the KT2442 K1-T6SS nor those of the IsoF *kib* system are known. However, that IsoF kills KT2442 may indicate

that *kib* is constitutively expressed and fires in a random fashion, while the K1-T6SS of KT2442 is only activated upon attack. This is reminiscent of the finding that a T6SS-negative *V. cholerae* strain is not killed by *P. aeruginosa*, whereas *V. cholerae* is efficiently killed in co-cultures with *P. aeruginosa* when both organisms contain a functional T6SS[43]. This would explain why all *kib* mutants of IsoF co-existed with KT2442. Additional work will be required to elucidate whether differences in the triggers or efficacies of the killing systems are responsible for the superior performance of the IsoF *kib* system. It is worth noting that IsoF killed many bacteria known to use T6SSs for interbacterial killing (Extended Data Fig. 10).

We demonstrated that *P. putida* IsoF has an unprecedented ability to invade and replace an established biofilm in a *kib*-dependent manner. In addition to *kib*, the production of the biosurfactant putisolvin, which enables the strain to swarm over semisolid surfaces, may contribute to the invasion competence of IsoF[45,46]. When mixed biofilms were grown in flow cells, killed cells were removed by the shear forces of the nutrient flow. The freed space was then occupied by IsoF, which eventually led to the replacement of the pre-established biofilm. Of note, putisolvin can effectively disperse biofilms grown on polyvinyl chloride and glass surfaces[45]. It will therefore be an interesting line of future research to elucidate the role of this biosurfactant in the removal of dead cells or the translocation of IsoF cells into the freed spaces in established biofilms.

We demonstrate that IsoF not only antagonizes several economically relevant phytopathogens but also protects tomato plants from *R. solanacearum* (Figs. 5 and 6). The presence of microbes secreting bacteriocins, antifungals or antibiotics in the rhizosphere is well known to be an effective strategy to suppress plant pathogens[15,47,48]. This study adds biofilm invasion through contact-dependent killing to the list of bacterial biocontrol functions. Given that a major limitation in biocontrol applications is that inoculants are unable to establish themselves in the environment, IsoF, which utilizes *kib* for attack as well as for defence, could be harnessed for eco-friendly farming strategies.

## Methods

**Bacterial growth conditions and media.** Bacterial strains used in this study are listed in Supplementary Table 1. Most bacterial overnight cultures were grown in Lysogeny Broth (LB, Difco, 240210) at 30 °C (*Pseudomonas* species) or at 37 °C (*Escherichia coli*). *R. solanacearum*, *Pectobacterium carotovorum*, *Pseudomonas syringae* overnight cultures and experiments were done in LB media without salt (LB−) at 30 °C. All other experiments were performed in AB medium[49] supplemented with 10 mM sodium citrate (indicated as ABC medium). If indicated, ABC was supplemented with 4 μg ml⁻¹ propidium iodide (PI, Thermo Fisher, P3566). Additionally, ABC was supplemented with 0.2% casamino acids if indicated (ABCAS). For selection of *Pseudomonas* mutants or transconjugants, *Pseudomonas* isolation agar (PIA, Difco, 292710) was used. If required, antibiotics were added at the following final concentrations: for *E. coli*: 100 μg ml⁻¹ ampicillin, 25 μg ml⁻¹ kanamycin (Km), 10 μg ml⁻¹ gentamycin (Gm), 10 μg ml⁻¹ tetracycline; for *Pseudomonas* species: 75 or 100 μg ml⁻¹ kanamycin, 20 or 30 μg ml⁻¹ gentamycin, 20 μg ml⁻¹ tetracycline.

**Construction of fluorescently tagged strains.** The mini-Tn7 system[50] was employed to integrate the gene encoding red fluorescent protein (mCherry) or Gfp into the chromosome of the strains listed in Supplementary Table 1. Mini-Tn7 tagged strains were obtained by tri-parental mating using the donor strain *E. coli* S17-1 carrying pUCT18-mini-Tn7 and the helper plasmid pUX-BF13[51–53]. Briefly, overnight cultures of the recipient strain, the helper strain and the donor strain were washed with 0.9% NaCl and then mixed in a 1:2:2 ratio (recipient:helper:donor). The strains were inoculated on LB plates as 50 μl drops and incubated at 30 °C overnight. Bacteria were resuspended in 1 ml 0.9% NaCl and plated on media containing Gm. Plates were incubated overnight at 30 °C and fluorescent colonies were selected.

**Tn5 mutant library, screening and mutant identification.** The transposon mutant library of IsoF was generated using the mini-Tn5 delivery vector pUT/mini-Tn5 Km[54]. Approximately 40,000 independent transposon insertion mutants were obtained after conjugation. Aliquots of the library were saved and stored at −80 °C. To perform the screening, individual mutants were grown overnight in 100 μl LB on 96-well plates, then the cultures were gently combined with 100 μl of *P. aureofaciens*::mCherry. The mixed-inocula were transferred to ABC medium

agar plates using a 96-pin replicator. Approximately 5,000 single Tn5 mutants were independently co-inoculated with *P. aureofaciens*::mCherry and incubated for 24 h at 30 °C. Mixed bacterial colonies were examined by means of fluorescence microscopy where competitions that showed red fluorescence indicated Tn5 mutants defective in killing. Initial hits were validated by contact-dependent killing assays as described later. Identification of the Tn5 insertion mutants was done by arbitrary PCR as previously described[55]. After the second round of PCR, reactions were cleaned with the PCR purification kit (Qiagen, 28006) and sequenced. Sequences were analysed and compared with the genome of IsoF and with NCBI Blast.

**CDK assays.** Overnight cultures were adjusted to an optical density (OD)$_{600}$ of 1 and dilutions were made to determine the number of c.f.u.s of each competitor. For the CDK assays, competitors were mixed in a 1:1 c.f.u. ratio. ABC, ABCAS or LB medium was inoculated with 5 μl of mixed culture. To determine the bacterial population in the mixed macrocolonies, c.f.u.s were counted at 0 h and at 24 h. At 24 h, two macrocolonies were resuspended in 500 μl of 0.9% NaCl and serial dilutions were plated on PIA and PIA Gm, the latter to select tagged strains. For the macrocolony overlaying killing assay, IsoF::Gfp was first inoculated and incubated at 30 °C for 1 h, then KT2442 was inoculated to cover half of the IsoF colony. Fluorescence of the mono and mixed cultures was examined using a Leica M165 FC fluorescence stereomicroscope.

**Monolayer killing assays.** On a microscope slide, 8–9 mm Ø × 1 mm depth adhesive silicon isolators (Grace BioLabs, JTR8R-1.0) were attached and filled with 62 μl ABC with 0.7% agar supplemented with PI. The middle of the agar was inoculated with 1 μl of the bacterial 1:1 mixed culture (IsoF::Gfp:competitor strain). The cover slip was placed on top after the inoculant had dried and killing was monitored with a confocal laser scanning microscope (CLSM) every 15 min for about 3 h. A final time point was recorded after 18–22 h of incubation at r.t., including samples of the monocultures. Image acquisition was done using a CLSM (Leica TCS SPE, DM5500) equipped with a ×100/1.44 oil objective. Images were analysed with ImageJ[56].

**Construction of *P. putida* IsoF deletion mutants.** IsoF derivatives with single and triple gene deletions and the deletion of the *kib* cluster (49.5 kbp) were constructed using SceI-based mutagenesis as described in ref. [57]. First, the plasmid pGPI-SceI (which carries an I-SceI recognition site) was modified by cloning *tetAR*, which encodes a tetracycline efflux pump into the PstI restriction site to give pGPI-SceI::TetAR. Next, two homology regions flanking the region to be deleted were cloned into pGPI-SceI::TetAR. The plasmid was introduced via conjugation and integrated into the genome of *P. putida* IsoF by single homologous recombination, giving two copies of the homologous regions in the chromosome. The plasmid pDAI::Gm$^R$, which carries the I-SceI nuclease, was then conjugated into the single-crossover IsoF strain. The I-SceI nuclease produced a double-strand DNA break at its recognition site, linearizing the chromosome and requiring recombination for the survival of the cell. This occurred preferentially at the repeated homologous regions. For both conjugations, the pRK2013 helper plasmid was used to provide the genes encoding the conjugation machinery. Ex-conjugants were selected on PIA Gm plates and screened by PCR using the check primers. Colonies were patched on PIA and PIA Gm20 to select colonies from which the pDAI plasmid had been cured. All primers and restrictions enzymes used for cloning are listed in Supplementary Table 2.

**Construction of pBBR1MCS derivative plasmids.** For complementation of the Δ32-33 and Δ33 mutants, plasmids pBBR::32-33 and pBBR::33 were constructed. Additionally, pBBR::32 was constructed. In each case, the coding sequence plus the native promoter region was amplified using an IsoF cell lysate as a template and cloned into pBBR1MCS-2 using primers and restriction sites as listed in Supplementary Tables 1 and 2. *E. coli* MC1061 was transformed with the ligated vectors, which were then transferred into the IsoF deletion mutants by tri-parental mating using *E. coli* DH5α pRK2013 as the helper strain. Complementation was checked by colony PCR using primers listed in Supplementary Table 2.

**Expression of PisoF_02332 and PisoF_02333.** Proteins produced from the genes *PisoF_02332* and *PisoF_02333* cloned into pBBR1MCS-2 were analysed by SDS–PAGE on a 15% gel as previously described[58]. Overnight cultures were lysed by sonication and protein concentration was estimated using the Bradford method (Sigma-Aldrich, B6916-500ML).

**Comparative genomic analysis.** Identification of *Pseudomonas* strains carrying the T4BSS gene cluster elements was performed using NCBI BLAST. The online tool ICEfinder was used to determine the boundaries of IsoF's genomic island (GI). The region containing the IsoF GI was compared against the 11 other *Pseudomonas* strains carrying T4BSS elements using the MAUVE v2.4.0 alignment tool with default settings[59]. MultiGeneBlast v.1.1.14 was done using the 11 *Pseudomonas* strains, IsoF and 8 additional known species in which the T4BSS has been described[60]. Alignment and comparison were done using the following settings to search for tightly coupled operons: gene identity threshold, 30%; number of hits mapped, 1,000; maximum distance between the genes in a locus, 10 kb. Alignment

for the phylogenetic trees was performed with CLC Genomics alignment tool and then manually curated, and the trees were reconstructed with RaxMLGUI v2.0 with 100 bootstrap repetitions. The final tree figures were done using FigTree v1.4.4.

**Tn-seq methodology.** Transposon mutagenesis was performed by tri-parental conjugation. First, the resistance properties of the donor plasmid pLG99 (carrying a Tn23 transposon) were modified by cloning a kanamycin resistance gene into the AatII restriction site. Overnight cultures of the recipient strain *P. putida* IsoF, the helper strain *E. coli* DH5α pRK2013 and the donor strain *E. coli* CC118 λ-pir pLG99::Km were washed with 0.9% NaCl and then mixed in a 1:2:2 ratio (recipient:helper:donor). The conjugation was plated on LB plates in drops of 50 μl and incubated for 2 h at 37 °C, followed by incubation at 30 °C overnight. The mating drops were resuspended in 6 ml 0.9% NaCl and plated on PIA containing Km. Plates were incubated at 30 °C overnight and the resulting colonies were washed from the plate with LB supplemented with Km. The resuspended mutant library was then mixed with an equal amount of 50% glycerol and kept at −80 °C. From three independent conjugations with approximately 70 matings, an estimated 700,000 mutants were generated.

For the Tn-seq experiments, the pooled mutant library was first grown for 16 h in liquid ABC media supplemented with 0.2% rhamnose until stationary phase. The $OD_{600}$ was then adjusted to 0.05 for growth in liquid medium (condition 1), and to an $OD_{600}$ of 1 for growth on solid medium as a monoculture (condition 2) and on solid medium as a co-culture (condition 3). For the growth in liquid medium, the Tn library was incubated at 30 °C for 4.5 h with 220 r.p.m. shaking, then cells were collected and pelleted for DNA extraction. For treatments 2 and 3, 400 drops of 5 μl each of the bacterial culture were plated. For the mixed condition (condition 3), *P. aureofaciens* was co-inoculated with the Tn mutant library in a 1:1 c.f.u. ratio. Both plated conditions were incubated for 8.5 h at 30 °C. Cells were scraped from the plate with 0.9% NaCl and adjusted to an $OD_{600}$ of 2 before being pelleted and kept at −20 °C for later DNA extraction. DNA extraction was done using the bacterial genomic DNA kit (Sigma-Aldrich, NA2110-1KT). All sequencing steps were performed using the previously described circle method[61], with several modifications described in ref. [62].

**Tn-seq data analysis and bioinformatics.** The Illumina sequencing reads were trimmed using Trimmomatic-0.32 (leading, 30; trailing, 30; slidingwindow, 4:20; Minlen, 60)[63]. Adapter sequences were removed with Cutadapt v1.9[64]. Tn-Seq Explorer was used to analyse the resulting Tn-seq data[65]. NCBI protein (.ptt) and RNA (.rnt) table files were generated from the IsoF genbank file (.gbff) and provided as input to Tn-Seq Explorer to infer the coordinates of proteins and RNA coding regions. Trimmed reads were mapped to the chromosome using the Bowtie2 plugin of Tn-Seq Explorer (–very-sensitive- command)[66]. A sequence alignment map (SAM) file was produced. For each dataset, the subsequent SAM generated with Bowtie2 was evaluated by Tn-Seq Explorer to assess essentiality. In the analysis, transposons mapping within 5% of the start codon and 20% of the stop codon were excluded. An estimated cut-off UID (unique insertion density) was established to separate essential from non-essential genes[65]. This was done by dividing the number of unique insertions by the gene length, resulting in a UID for that specific gene. The number of genes with the given insertion density versus the insertion density per bp was represented in a plot, usually showing a bimodal distribution. Here the genes with low or no-insertions appeared on the left side of the plot. The point where the plot rises again indicates the threshold for genes that can tolerate transposon insertions[61]. This point indicates the cut-off for essential genes, which was set for each growth condition: liquid, 0.013; plate, 0.011; mixed, 0.014 (UID). Genes showing higher UID values were considered non-essential since high number of transposon insertions was detected per gene. Mapped transposon insertions are listed in Supplementary Figs. 2–4.

**Flow-cell biofilms, microscopy and image analysis.** Biofilms were grown in a flow-cell system[67,68] with continuous flow of liquid AB medium supplemented with 0.1 mM sodium citrate at a rate of 0.2 mm s⁻¹ using a Watson-Marlow 205S peristaltic pump. Briefly, the flow-cell chambers were inoculated with *P. putida* cultures at an $OD_{600}$ of 0.1 and biofilm development was followed every 24 h for up to 5 d. For the competition experiment, the strain inoculated on top of the pre-established biofilm was adjusted to an $OD_{600}$ of 0.5. For the two-species biofilm, strains were mixed in a 1:1 ratio and cultivated for up to 48 h. Shortly before 40 h of cultivation, PI was added. Photomicrographs were taken every 24 h with a CLSM (Leica TCS SPE, DM5500) equipped with a 63 ×1.3 oil objective. Monolayer killing assays were imaged with a 100 ×1.44 oil objective. Images were analysed with the Leica Application Suite, the Imaris v9.6.0 software package (Bitplane) and with ImageJ[56].

**Plant assay on MS plates.** Micro-Tom *Solanum lycopersicum* L. seeds (Tuinplus bv. Heerenveen, Holland) were surface-sterilized with 1% sodium hypochlorite solution for 10 min, washed 4 times with sterile distilled water (dH₂O) and placed at 4 °C for 2 d in the dark. Seeds were then sown on 0.8% water agar plates and kept at 30 °C for 2 d in the dark. Germinated seeds were incubated at 22 °C in long-day conditions (16 h, 100 μmol m⁻² s⁻¹ photon flux, 20 °C light and 8 h, 18 °C dark regime at 60% relative humidity) and seedlings were further grown for

7–8 d until lateral root emergence. Seedlings were injured twice with a 0.4 mm diameter needle at the root-shoot junction. The roots of the seedlings were then submerged for 10 s each in a bacterial suspension set to a final $OD_{600}$ of 0.5 in 1 mM MgSO₄. Inoculated seedlings were next placed on half strength Murashige and Skoog (MS) medium (Sigma-Aldrich, M5519-50L) with 1.5% agar and grown for 22 d under the long-day conditions indicated above. Each tomato plant root was washed and sonicated and roots were then placed in an Eppendorf tube with 750 μl of 1 mM MgSO₄[69]. The washing step consisted of shaking the tube for 15 min at 160 r.p.m., followed by sonication for 15 min. The obtained cell suspensions were serial-diluted to allow for c.f.u. quantification. Principal component analysis and hierarchical clustering were performed on individual root weight, shoot area and chlorophyll values. Unit variance scaling was applied and the single value decomposition with imputation used to calculate principal components. Prediction ellipses were used to display the 95% confidence intervals. Root parameters were clustered using correlation distance and complete linkage, and plant samples were clustered using Euclidean distance and complete linkage (n = 180). Three independent biological replicates were performed, with a minimum of 28 plants for each treatment. Shoot area and chlorophyll estimations were obtained from calibrated RGB photographs using ImageJ[70], adapting previous methods[71]. Essentially, individual blue RGB values were extracted from blue channel-thresholded plant pictures and normalized to total RGB. Normalized green and red channel values were used to calculate a greenness index (Greendex = 4G – 3R). Greendex values and acetone-extracted total chlorophyll per shoot weight of single infected or healthy tomato plantlets were linearly correlated ($R^2 = 0.7373$, n = 21)[72]. Principal component analysis and hierarchical clustering data visualization were done using ClustVis Webtool (https://bio.tools/clustvis).

**Infection of tomato seedlings by soil drenching.** The protocol from ref. [73] was used with the following modifications: bacterial dilutions were done with room temperature tap water and high-volume inocula were prepared ($OD_{600}$ of 0.1, 0.01 and 0.001) of *R. solanacearum*, approximately equivalent to 10⁸ c.f.u. ml⁻¹, 10⁷ c.f.u. ml⁻¹ and 10⁶ c.f.u. ml⁻¹ and ($OD_{600}$ of 1) of IsoF/ΔT4B, approximately equivalent to 10⁷ c.f.u. ml⁻¹. For the trial assay with *R. solanacearum* only, non-sterile soil was drenched with different c.f.u. amounts (see above) and uninjured and injured seedlings were tested (n = 1, 20 plants per treatment). The dosage of 10⁸ c.f.u. ml⁻¹ of *R. solanacearum* on injured seedlings was the treatment chosen as it showed the strongest effect on the plantlets where ~66% of the plants did not progress beyond the 2-leaf stage. Eight-day-old tomato seedlings were slightly injured at the root-stem junction. Before potting the seedlings, the non-sterile soil (used 7 × 7 × 6 cm pots, Desch Plantpak, 100003) was drenched with 50 ml of the following bacterial suspensions: monocultures of *Ralstonia*::Gfp, IsoF::mCherry and ΔT4B::mCherry; and mixed cultures of *Ralstonia*::Gfp + IsoF::mCherry and *Ralstonia*::Gfp + ΔT4B::mCherry. In the mixed cultures, the $OD_{600}$ was adjusted so that the final concentration of both bacterial competitors was 10⁸ c.f.u. ml⁻¹. Plantlets were grown in a light chamber for 20 d and bacterial c.f.u.s were recovered from the rhizosphere. Three independent biological replicates were carried out, with a total of 41 plants per treatment (number of seedlings per replicate: 1st = 20, 2nd = 11, 3rd = 10). Leaf area was measured using RGB value-thresholded images with ImageJ.

**Statistics and reproducibility.** Statistical significance was assessed by appropriate tests as stated in figure legends. Analyses were performed using GraphPad Prism v8.4.1, with $P < 0.05$ considered significant. Student's $t$-tests were employed and asterisks indicate the level of significance: *$P < 0.05$, **$P < 0.01$, ***$P < 0.001$, ****$P < 0.0001$. When required, significantly different means in analysis of variance (ANOVA) with Tukey's multiple comparisons ($P < 0.05$) are indicated by different letters. No statistical method was used to predetermine sample size and investigators were not blinded to allocation during experiments and outcome assessment. The experiments were not randomized and no data were excluded from the analyses.

**Reporting summary.** Further information on research design is available in the Nature Research Reporting Summary linked to this article.

## Data availability
The genome sequence of IsoF has been deposited in NCBI under accession number CP072013. FASTQ files generated from the Illumina MiSeq platform are publicly available at the NCBI short reads archive (SRA) under BioProject PRJNA730700. Individual datasets have the following accession numbers: Liquid, SRR14612110; Plate, SRR14612109; and Mixed, SRR14612108. Source data are provided with this paper.

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

## Acknowledgements

This work was funded in part by Swiss National Science Foundation grants CRSII5_186410 and 310030_192800. K.A. was funded by the Gebert Rüf Stiftung grant GRS-076/79. We thank C. Aguilar for assistance in constructing the IsoF mini-Tn*5* library and A. Vitale for help with the TnSeq; C. Fabbri, J. Steger and E. Leu for technical support; Y. Chen and S. Gualdi for helpful discussions.

## Author contributions

G.P.-M., G.C.-O. and L.E. designed the overall experimental plan for the study. G.P.-M. performed the majority of the experiments presented and wrote the manuscript with input from all authors. G.C.-O. contributed to project management and performed the Tn*5* library experiments, the initial bacterial competition experiments and the flow-cell biofilm experiments. M.P.-C. contributed to the analysis of the TnSeq library, bioinformatic analyses, and the sequencing and annotation of the genome of IsoF. K.A. supported all molecular microbiology experiments and contributed to writing the manuscript. A.B. contributed to the design of the plant experiment and performed the principal component analysis, hierarchical clustering and related analyses. L.E. contributed to project management and writing of the manuscript.

## Competing interests

The authors declare no competing interests.

## Additional information

**Extended data** is available for this paper at https://doi.org/10.1038/s41564-022-01209-6.

**Correspondence and requests for materials** should be addressed to Leo Eberl.

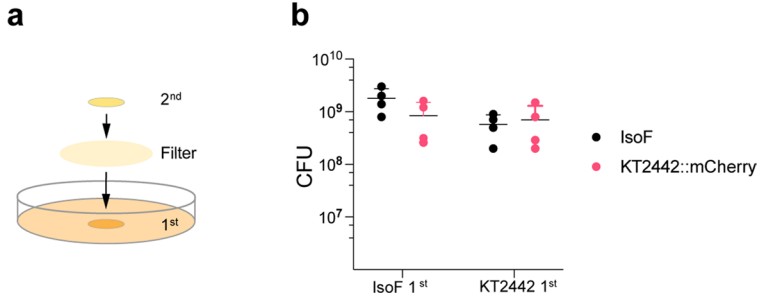

**Extended Data Fig. 1 | Inoculation of KT2442::mCherry and IsoF separated by a filter. a**, IsoF was inoculated on an ABC plate, a filter was placed atop the bacteria, which was subsequently inoculated with KT2442 or *vice versa*. **b**, CFUs were determined after 24 h of incubation. Data are mean ± s.d. from four independent replicates (n = 4).

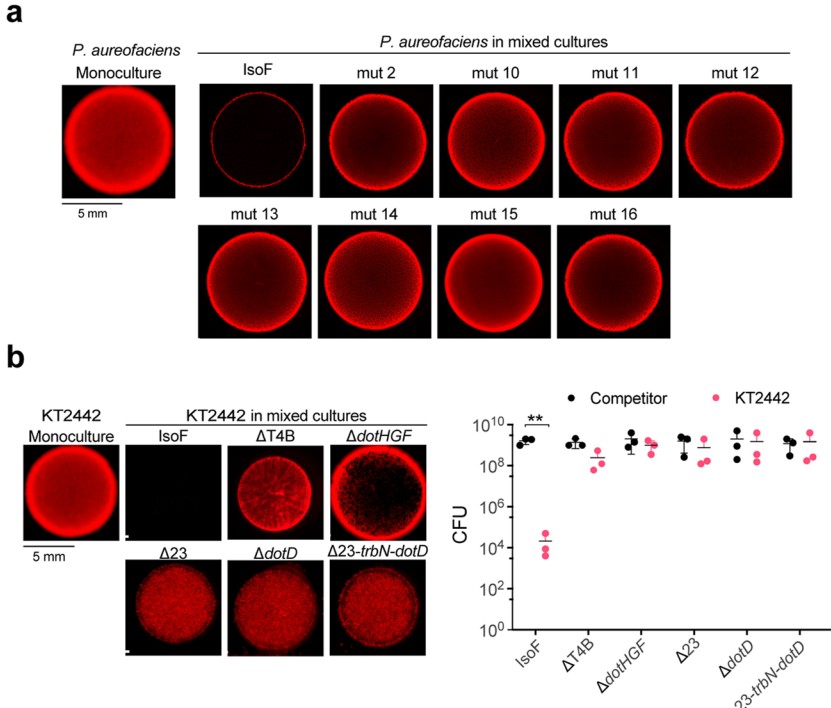

**Extended Data Fig. 2 | Contact-dependent killing (CDK) assays. a**, IsoF mini-Tn5-insertion mutants versus *P. aureofaciens*. The mCherry (red) signal indicates the survival of *P. aureofaciens*::mCherry in competition with the mini-Tn5 insertion mutants. **b**, CDK of defined mutants against KT2442. The wild-type strain and the deletion mutants Δ*dotHGF*, Δ23, Δ*dotD*, Δ23-*trbN-dotD* and ΔT4B were co-inoculated with KT2442 tagged with mCherry. CFUs were determined after 24 h of incubation. Data are mean ± s.d. of three independent biological replicates (n = 3). Unpaired t-test, **\*\*P < 0.01; \*\*\*P < 0.001.** Representative pictures of at least 3 replicates are shown. Scale bar, 5 mm.

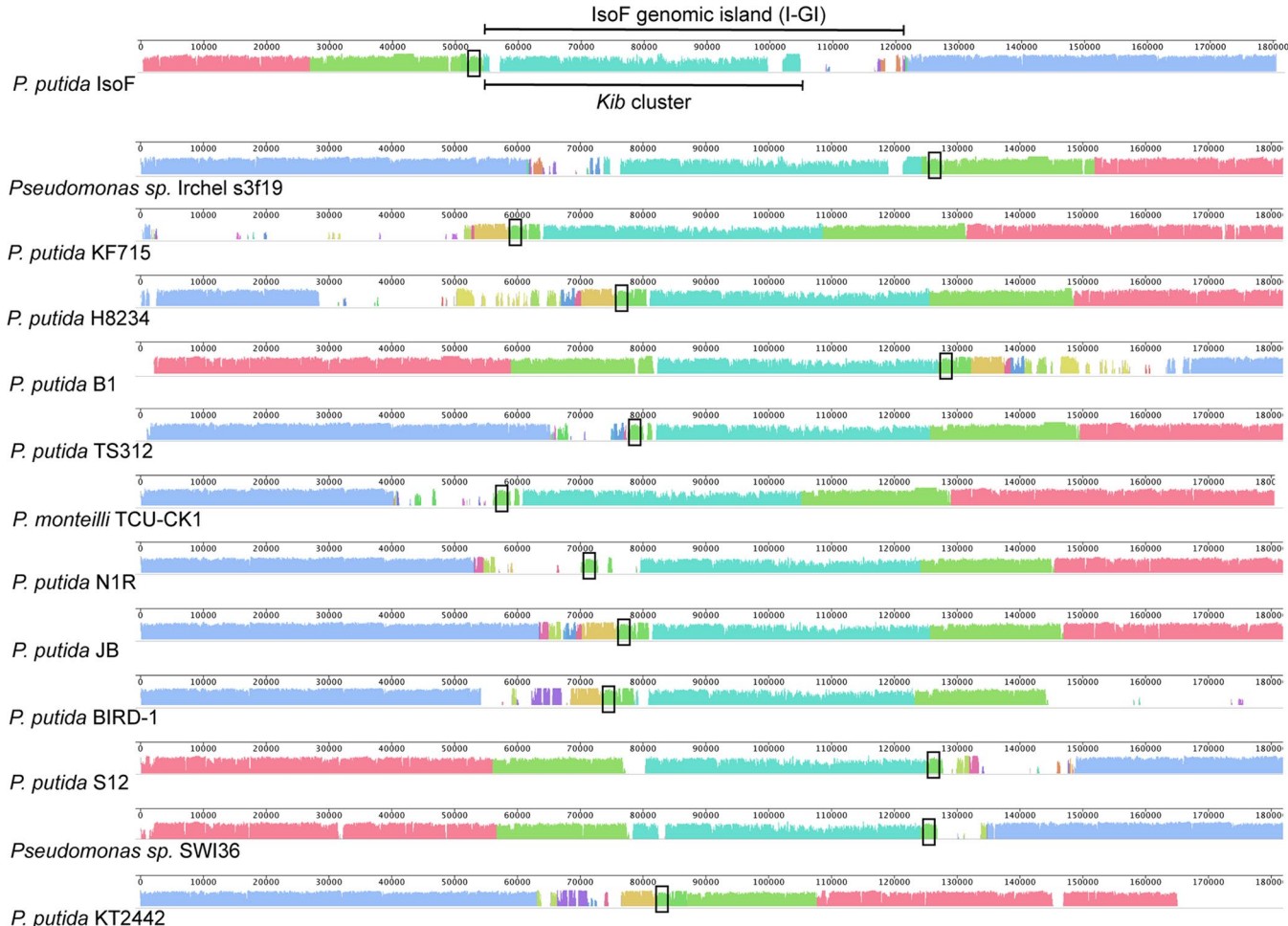

**Extended Data Fig. 3 | Multiple genome alignment of *kib* loci.** The *kib* regions from eleven *Pseudomonas* strains were compared and aligned with the IsoF-GI region using the Mauve software[1]. *P. putida* KT2440 (AE015451.2) was used as a reference strain not carrying this genomic element. Regions with identical colors represent local collinear blocks (LCB) of conserved DNA sequence. Low identity regions are shown without LCB. The *kib* gene cluster regions are indicated in light blue. The rectangular box (black) indicates the same position in a homologous region that is shared by all strains. The strains' accession numbers are: *Pseudomonas* sp. Irchel (NZ_FYDW01000002), *P. putida* KF715 (AP015029.1), *P. putida* H8234 (CP005976.1), *P. putida* B1 (NZ_CP022560.1), *P. putida* TS312 (NZ_AP022324.1), *P. monteilli* TC1-CK1 (NZ_CP040324.1), *P. putida* N1R (NZ_LT707061.1), *P. putida* JB (NZ_CP016212.1), *P. putida* BIRD-1 (CP002290.1), *P. putida* S12 (NZ_CP009974.1), *Pseudomonas* sp. SWI36 (NZ_CP026675.1).

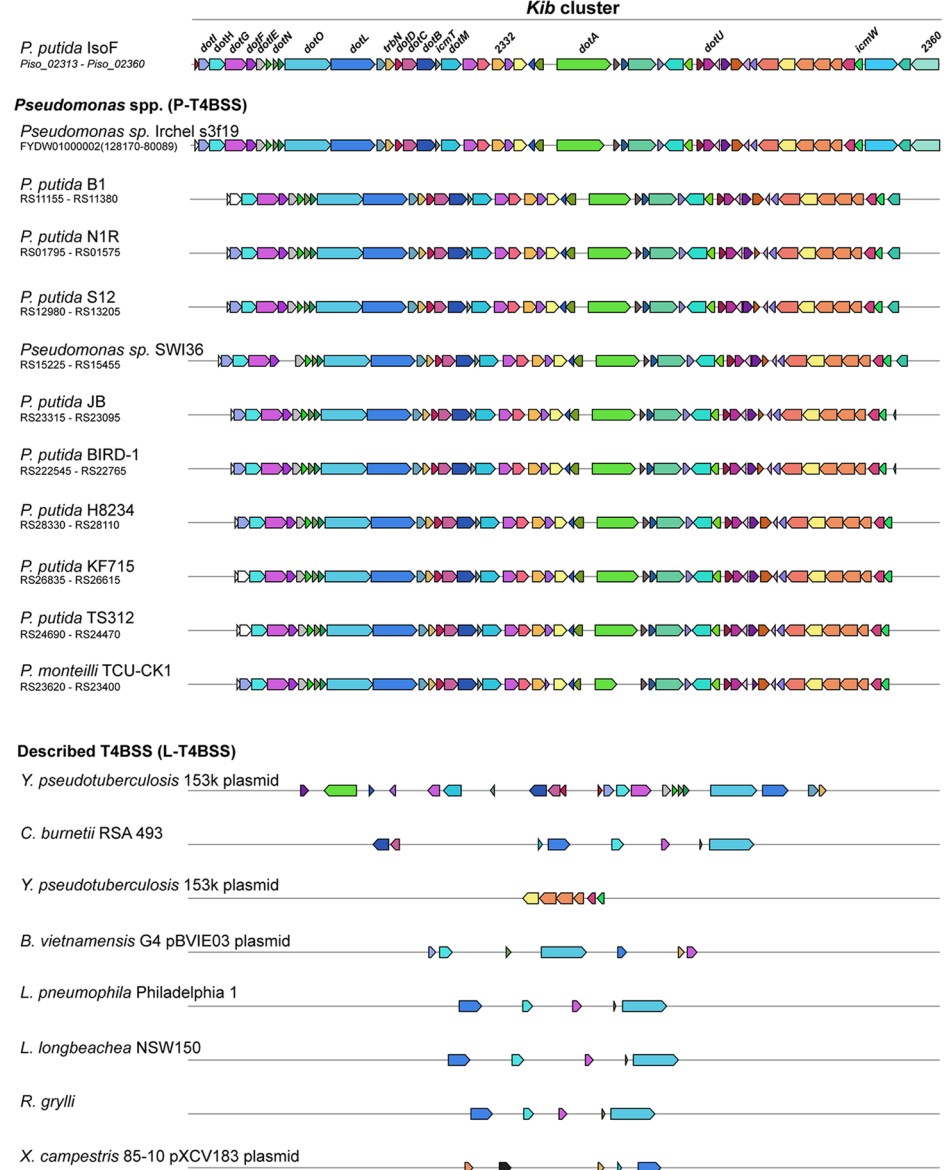

**Extended Data Fig. 4 | MultiGeneBlast alignment of IsoF *kib* genes with homologs of *Pseudomonas* and other T4BSS-encoding bacteria.** MultiGeneBlast Alignment[2] of homologous regions from eleven *Pseudomonas* strains (P-T4BSS) and eight T4BSS-encoding bacteria as described by Nagai and Kubori, 2011[3] (L-T4BSS). Arrows represent genes and the color conservation indicates homology > 30 %. Gene identity threshold: 30%, hits mapped: 1000. Maximum distance between the genes in a locus is 10 kb.

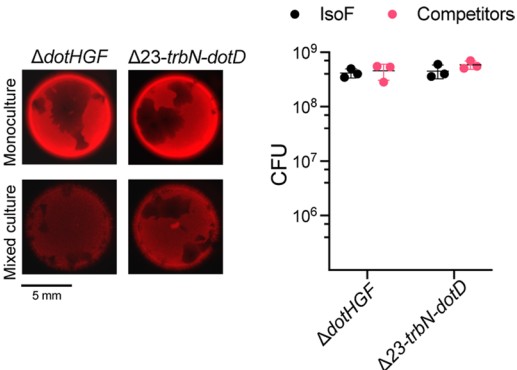

**Extended Data Fig. 5 | CDK between the IsoF wildtype and mutants Δ*dotHGF*::mCherry and Δ23-*trbN-dotD*::mCherry.** Both mutant strains survived after 24 h of coincubation. CFUs are mean ± s.d. from three independent replicates (n = 3). Representative images are shown. Scale bar, 5 mm.

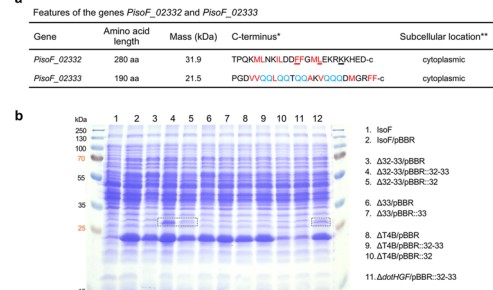

**Extended Data Fig. 6 | Ectopic expression of PisoF_02332 and PisoF_02333 from plasmids pBRR::32, pBRR::33 and pBRR::32-33 in different genetic backgrounds. a**, The length, expected mass (kDa) and predicted subcellular location of the effector-immunity pair proteins are indicated. The C-termini of both proteins were examined for conserved recognition sequences such as hydrophobic residues (red), EExxE and FxxxLxxxK domains (underlined)[4–7]. * The last 25 aa residues of the c-terminus region are indicated; ** LocTREE and PSORTb web tools were employed. An unusual glutamine-rich domain (blue) was observed in the C-terminal region of the effector PisoF_02333. **b**, SDS-PAGE of cell lysates. Proteins were separated on 15% polyacrylamide gels and stained with Coomassie Blue R-250. Dotted squares indicate PisoF_02332 (31.9 kDa). PisoF_02333 could not be detected in any of the samples. Black arrow indicates plasmid-encoded proteins.

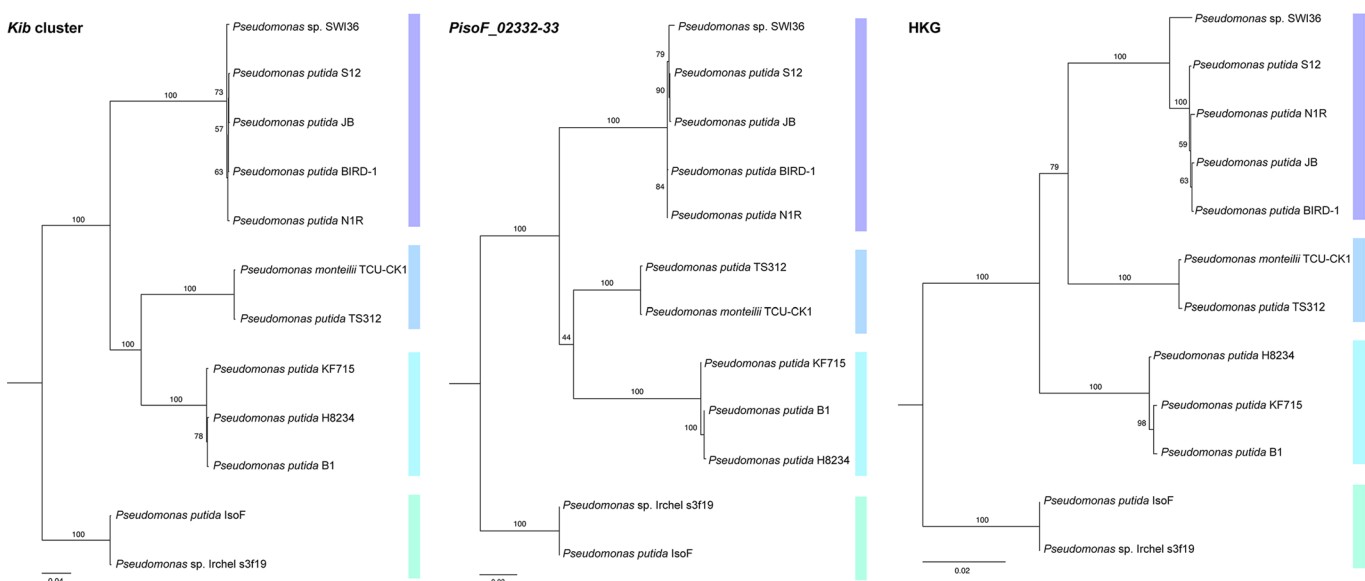

**Extended Data Fig. 7 | Phylogenetic trees of all orthologs of the *kib* cluster, the E-I genes *PisoF_02332-33*, and eight housekeeping genes (HKG).** The *kib* cluster tree is based on the T4BSS orthologs *Piso_02313* to *icmW* and the HKG tree based on eight concatenated genes of the listed species (*gyrB, rpoD, 16 s, argS, dnaN, dnaQ, gltA, rpoB*). Coloured bars indicate four distinct lineages. The bootstrap values are displayed on the trees.

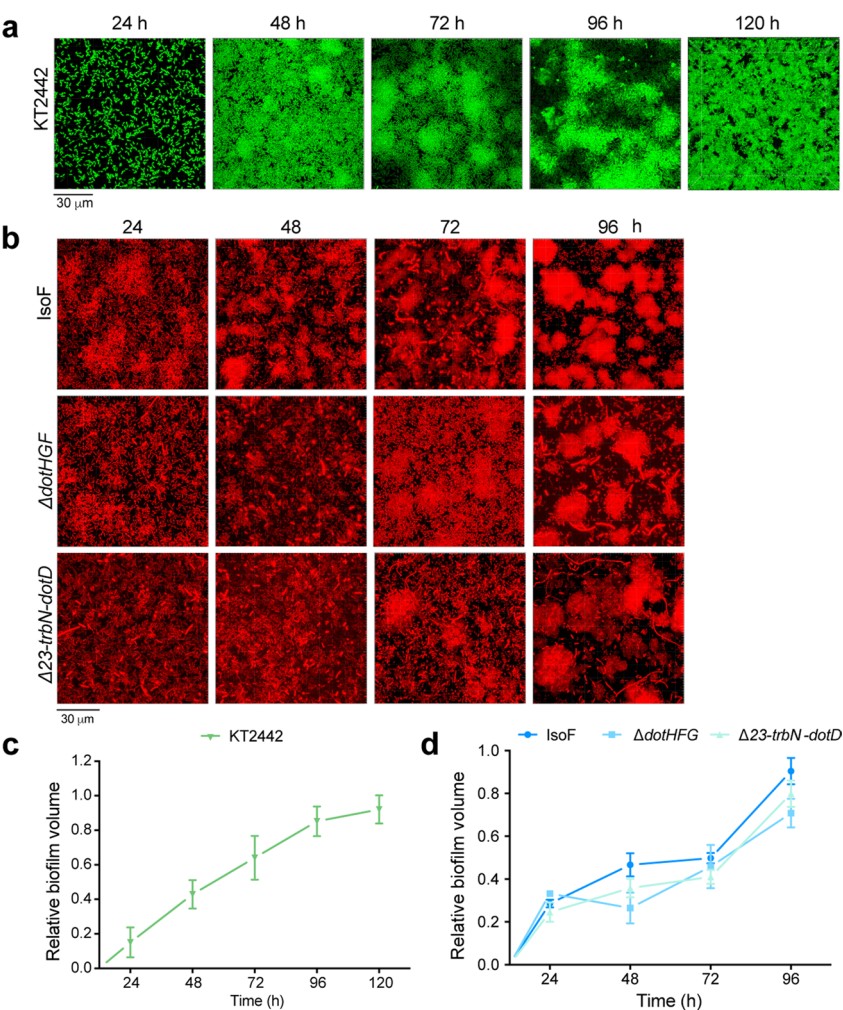

**Extended Data Fig. 8 | Biofilm development of KT2442 (green), IsoF, Δ*dotHGF* and Δ23-*trbN-dotD* (red). a**, Representative pictures of KT2442 biofilm development over 120 h. **b**, Representative images of biofilm development over 96 h. **c,d**, Quantification of biomass (relative volume) increase over time. Biofilms were visualized by CLSM using a 63 ×1.3 oil objective. Biomass (volume) of the biofilm was quantified by the Imaris software (Bitplane). Data are mean ± s.d. of a minimum of two biological replicates (n = 3). Scale bar, 30 μm.

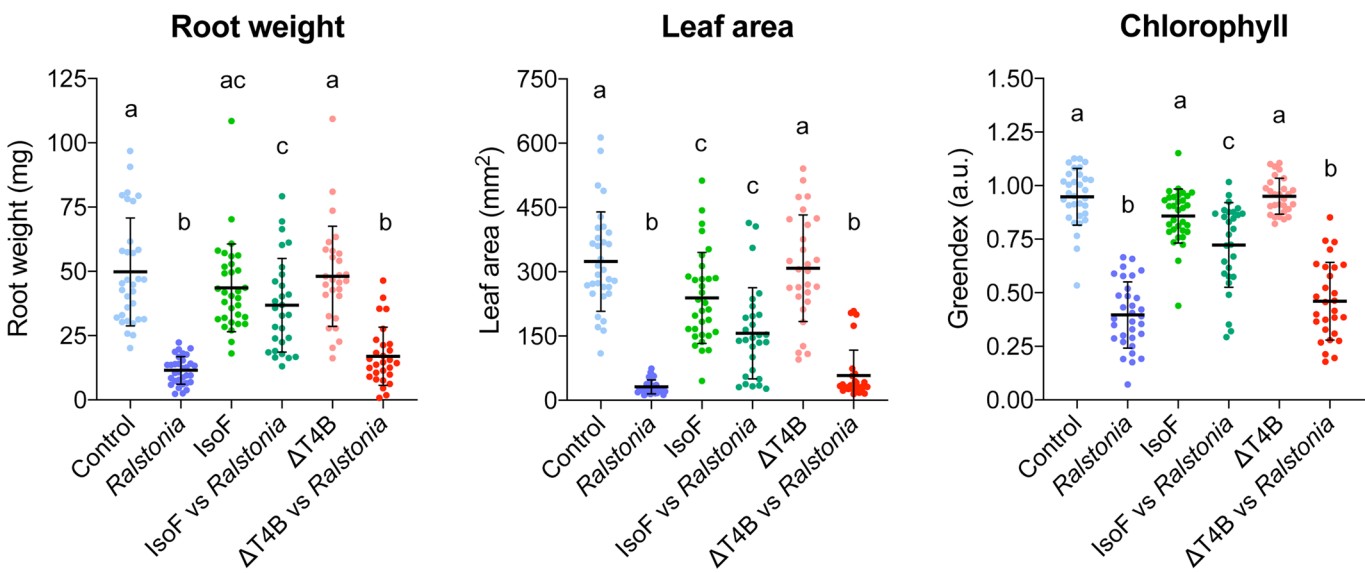

**Extended Data Fig. 9 | Root weight, leaf area and chlorophyll content (Greendex) of tomato plants grown on MS plates.** Treatments labelled with the same letter are not significantly different. ANOVA, Tukey's test, $P < 0.05$. Error bars show the mean ± s.d. ($n = 3$).

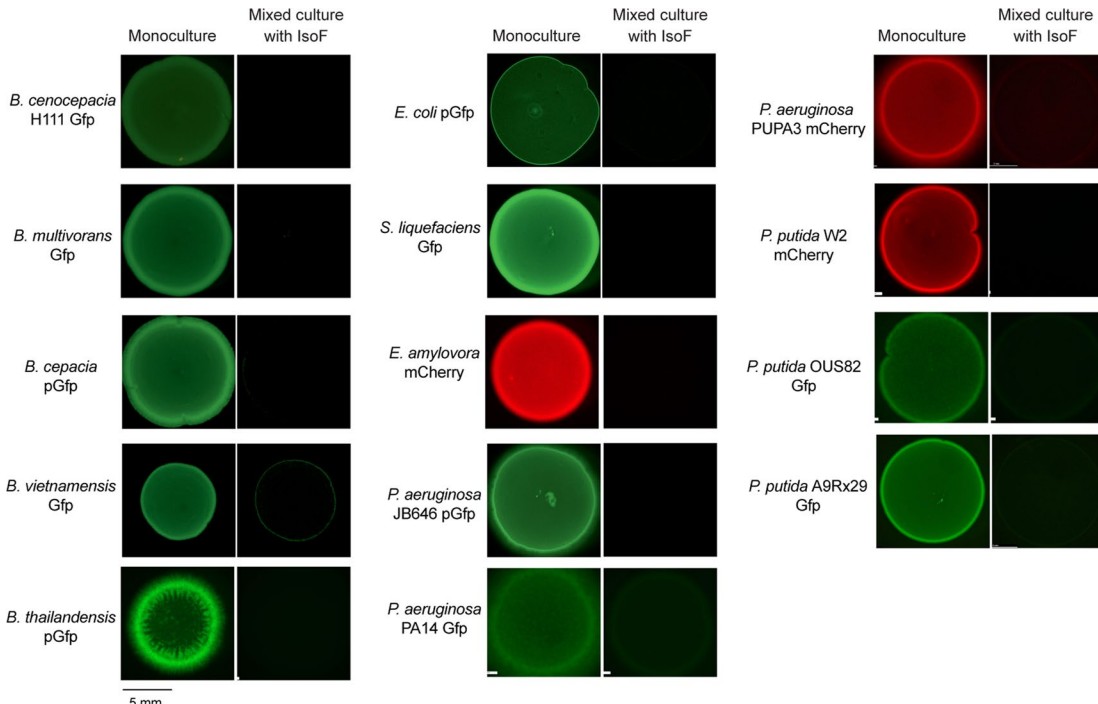

**Extended Data Fig. 10 | IsoF kills a wide range of Gram-negative bacteria.** Various fluorescently tagged bacteria were competed against the IsoF wildtype. Fluorescence images of the mono- and mixed-cultures after 24 h of incubation are shown. Lack of fluorescence indicates that IsoF killed the bacterial strain. Several bacteria that were shown to use T6SSs for interbacterial killing were killed by IsoF, including *P. aeruginosa* PA14[8], *B. cenocepacia* H111[9], *P. syringae*[10], *P. chlororaphis*[11], *P. fluorescens*[12], *P. carotovorum*[13], *E. amylovora*[14] and *B. thailandensis*[15]. Scale bar, 5 mm.

# Reporting Summary

## Statistics

For all statistical analyses, confirm that the following items are present in the figure legend, table legend, main text, or Methods section.

| n/a | Confirmed | |
|---|---|---|
| ☐ | ☒ | The exact sample size (*n*) for each experimental group/condition, given as a discrete number and unit of measurement |
| ☐ | ☒ | A statement on whether measurements were taken from distinct samples or whether the same sample was measured repeatedly |
| ☐ | ☒ | The statistical test(s) used AND whether they are one- or two-sided *Only common tests should be described solely by name; describe more complex techniques in the Methods section.* |
| ☒ | ☐ | A description of all covariates tested |
| ☐ | ☒ | A description of any assumptions or corrections, such as tests of normality and adjustment for multiple comparisons |
| ☐ | ☒ | A full description of the statistical parameters including central tendency (e.g. means) or other basic estimates (e.g. regression coefficient) AND variation (e.g. standard deviation) or associated estimates of uncertainty (e.g. confidence intervals) |
| ☐ | ☒ | For null hypothesis testing, the test statistic (e.g. *F*, *t*, *r*) with confidence intervals, effect sizes, degrees of freedom and *P* value noted *Give P values as exact values whenever suitable.* |
| ☒ | ☐ | For Bayesian analysis, information on the choice of priors and Markov chain Monte Carlo settings |
| ☒ | ☐ | For hierarchical and complex designs, identification of the appropriate level for tests and full reporting of outcomes |
| ☒ | ☐ | Estimates of effect sizes (e.g. Cohen's *d*, Pearson's *r*), indicating how they were calculated |

*Our web collection on statistics for biologists contains articles on many of the points above.*

## Software and code

Policy information about availability of computer code

| Data collection | Leica Application Suite LAS V4.5 and Leica Application Suite LAS AF 2.7.3 (Leica Microsystems, Germany) |
|---|---|
| Data analysis | IMARIS v9.6.0 (Bitplane, Switzerland). PRISM v8.4.1 (GraphPad Software). ImageJ v1.53f51 (Schneider et al., 2012). ClustVis Webtool (Metsalu et al., 2015). MAUVE v2.4.0 (Darling et al., 2004). MultiGeneBlast v1.1..14 (Medema et al., 2013). RaxmlGUI v2.0 (Edler et al 2021) |

For manuscripts utilizing custom algorithms or software that are central to the research but not yet described in published literature, software must be made available to editors and reviewers. We strongly encourage code deposition in a community repository (e.g. GitHub). See the Nature Portfolio guidelines for submitting code & software for further information.

## Data

Policy information about availability of data

All manuscripts must include a data availability statement. This statement should provide the following information, where applicable:
- Accession codes, unique identifiers, or web links for publicly available datasets
- A description of any restrictions on data availability
- For clinical datasets or third party data, please ensure that the statement adheres to our policy

The genome sequence of IsoF has been deposited in NCBI under the accession number CP072013. FASTQ files generated from the Illumina MiSeq platform are publicly available at the NCBI short reads archive (SRA) under the BioProject:   PRJNA730700. Individual datasets have following accession numbers: Liquid: SRR14612110, Plate: SRR14612109 and Mixed: SRR14612108.

# Field-specific reporting

Please select the one below that is the best fit for your research. If you are not sure, read the appropriate sections before making your selection.

☒ Life sciences ☐ Behavioural & social sciences ☐ Ecological, evolutionary & environmental sciences

For a reference copy of the document with all sections, see nature.com/documents/nr-reporting-summary-flat.pdf

# Life sciences study design

All studies must disclose on these points even when the disclosure is negative.

| | |
|---|---|
| Sample size | We did not perform a sample-size calculation but made sure that a sufficient number of replicates were taken following previous experience with these types of experimental data. Furthermore, the detailed statistical analysis of the data generated in this study indicated a high level of reproducibility. Specifically, all competition experiments were performed at least as biological triplicates (exact number of replicates indicated by dots) and each replicate measure comprised of 2 technical replicates. In the monolayer competition experiments, the number of dead cells was determined by at least 3 technical replicates. For the flow cell biofilm experiments, up to 3 biological replicates were carried out and each replicate measurement comprised 3 technical replicates. In planta experiments were performed in biological triplicate, using a minimum of 8 plants per treatment for each replicate. |
| Data exclusions | No data were excluded from the analysis. |
| Replication | Experiments were performed at least in triplicate. The Tn-seq analysis was carried out only once for each of the three growth regimes, as we used a highly saturated transposon library containing more than 700.000 insertion mutants, i.e. an insertion every 8 bp. By using a large mutant number as inoculum we made sure that no bottlenecks were created in our experiments, which otherwise may have create a bias. It is also noteworthy that all three growth regimes identified that same genes and thus can be considered biological replicates. |
| Randomization | Randomization is not relevant for our study that investigated interactions between bacteria and bacteria and their host plant. |
| Blinding | Blinding is not relevant for our study that investigated interactions between bacteria and bacteria with their host plant. |

# Reporting for specific materials, systems and methods

We require information from authors about some types of materials, experimental systems and methods used in many studies. Here, indicate whether each material, system or method listed is relevant to your study. If you are not sure if a list item applies to your research, read the appropriate section before selecting a response.

## Materials & experimental systems

| n/a | Involved in the study |
|---|---|
| ☒ | ☐ Antibodies |
| ☒ | ☐ Eukaryotic cell lines |
| ☒ | ☐ Palaeontology and archaeology |
| ☒ | ☐ Animals and other organisms |
| ☒ | ☐ Human research participants |
| ☒ | ☐ Clinical data |
| ☒ | ☐ Dual use research of concern |

## Methods

| n/a | Involved in the study |
|---|---|
| ☒ | ☐ ChIP-seq |
| ☒ | ☐ Flow cytometry |
| ☒ | ☐ MRI-based neuroimaging |

