## [Peer Review File. · Nature Microbiology]

Peer Review Information

Journal: Nature Microbiology

Manuscript Title: Pseudomonas putida mediates bacterial killing, biofilm invasion and biocontrol with a type IVB secretion system

Corresponding author name(s): Leo Eberl

Reviewer Comments & Decisions:

Decision Letter, initial version:

Dear Professor Eberl,

Thank you for your patience while your manuscript "A type IVB secretion system adapted for bacterial killing, biofilm invasion and biocontrol" was under peer-review at Nature Microbiology. It has now been seen by 3 referees, whose expertise and comments you will find at the end of this email. Although they find your work of some potential interest, they have raised a number of concerns that will need to be addressed before we can consider publication of the work in Nature Microbiology.

In particular, you will see that referee #1 raise concerns regarding the plant experiment and asks you to use a more biologically relevant approach, and suggests you to tone down some statements and improve the data presentation and description. Referee #2 raises issues regarding the lack of data availability. Referee #3 mainly comments on the need for more mechanistic insight to strengthen the manuscript.

Should further experimental data allow you to address these criticisms, we would be happy to look at a revised manuscript.

Please include a data availability statement as a separate section after Methods but before references, under the heading "Data Availability". This section should inform readers about the availability of the data used to support the conclusions of your study. This information includes accession codes to public repositories (data banks for protein, DNA or RNA sequences, microarray, proteomics data etc...), references to source data published alongside the paper, unique identifiers such as URLs to data repository entries, or data set DOIs, and any other statement about data availability. At a minimum, you should include the following statement: "The data that support the findings of this study are

2available from the corresponding author upon request", mentioning any restrictions on availability. If DOIs are provided, we also strongly encourage including these in the Reference list (authors, title, publisher (repository name), identifier, year). For more guidance on how to write this section please see:

<http://www.nature.com/authors/policies/data/data-availability-statements-data-citations.pdf>

* If you have not done so already we suggest that you begin to revise your manuscript so that it conforms to our Article format instructions at <http://www.nature.com/nmicrobiol/info/final-submission>. Refer also to any guidelines provided in this letter.

When submitting the revised version of your manuscript, please pay close attention to our [href="https://www.nature.com/nature-research/editorial-policies/image-integrity">Digital Image Integrity Guidelines. and to the following points below:](https://www.nature.com/nature-research/editorial-policies/image-integrity)

{redacted}

Note: This url links to your confidential homepage and associated information about manuscripts you may have submitted or be reviewing for us. If you wish to forward this e-mail to co-authors, please delete this link to your homepage first.

Nature Microbiology is committed to improving transparency in authorship. As part of our efforts in this direction, we are now requesting that all authors identified as 'corresponding author' on published papers create and link their Open Researcher and Contributor Identifier (ORCID) with their account on the Manuscript Tracking System (MTS), prior to acceptance. This applies to primary research papers only. ORCID helps the scientific community achieve unambiguous attribution of all scholarly contributions. You can create and link your ORCID from the home page of the MTS by clicking on 'Modify my Springer Nature account'. For more information please visit www.springernature.com/orcid.

If you wish to submit a suitably revised manuscript we would hope to receive it within 6 months. If you cannot send it within this time, please let us know. We will be happy to consider your revision, even if a similar study has been accepted for publication at Nature Microbiology or published elsewhere (up to a maximum of 6 months).

Yours sincerely,

{redacted}

Reviewer Expertise:

Referee #1: microbial ecology of plant bacteria/Pseudomonas

Referee #2: Type IV secretion systems

Referee #3: plant microbe signaling/QS

Reviewer Comments:

Reviewer #1 (Remarks to the Author):

This very interesting study has done a good job of describing a novel form of contact-dependent killing of bacteria. Contact-dependent killing schemes have become quite the rage in microbial ecology, and yet most have been associated with variants of type VI secretion systems. This study has shown a very novel method of delivering what appear to be toxic effectors using a variant of the type IVB secretion system. The study is quite solid, and the authors have employed a variety of powerful tools including Tn sequencing and various microscopy techniques to both demonstrate the contact dependence of the killing of various other bacteria by the *Pseudomonas putida* strain, and to identify the genetic loci associated with both the toxic effector as well as the immunity function. It is quite remarkable that this type IVB secretion system seems to be quite promiscuous in its ability to kill other bacteria, as they did not seem to find any taxa that were immune to its effect. They might want to comment further on the relative breadth of this killing compared to many other type VI systems that have been studied. As such, there are indeed important translational aspects of the study, as exemplified here in their demonstration that the biological control of bacterial wilt caused by *Ralstonia solanacearum* could be achieved using this strain capable of contact killing. The manuscript was quite

3well written, and the powerful and logical experimental design leading to the discovery of the system was well considered. I have only a few relatively minor comments about both the way certain findings should be described, a few conclusions that may be a bit overstated, as well as suggestions for how the demonstration of plant disease control could have been better assessed and described.

Specifics:

Lines 67-69. We are told here that a *Xanthomonas* strain had exhibited a type IV dependent killing of other bacterial strains - while it is only in a couple of sentences later that the authors note that there is more than one kind of type IV secretion system, and that there had been no prior demonstration of the type IVB system being used in antimicrobial activities. However, there was never a mention until much later in the discussion that the *Xanthomonas* example, was in fact, a type IVA secretion system. I feel that this point should have been made earlier, because I was bothered by the concern that this study might not have been as novel as they were suggesting, because they had not noted that this earlier example was not a type IVB system.

Line 90 and elsewhere. Here and in many locations throughout the manuscript, the authors have used the term "competition" to describe the interaction of IsoF with various other bacteria. At least in my mind, the term "competition" implies some sort of growth reduction that would be a result of the need to share certain nutrient resources with a neighbor, or perhaps an effect of the chemical environment by a neighbor etc. Given that they are demonstrating here that "competition" is actually the direct killing of their neighbor, I wonder if it would be cleaner to refer to "interaction" or "killing" or some other term that does not seem to have this nutrient competition connotation.

Line 97 misspelling: iodide

Line 112. I had to take a step back and try to figure out why the authors had decided to use *Pseudomonas aureofaciens* in the study discussed in this section because there had not been a previous mention that this particular species was in fact susceptible to the killing effect of strain IsoF. One had to dig into the results of Figure 1, and specifically recognize that *Pseudomonas aureofaciens* was one of the strains that was in fact susceptible to the killing by IsoF. I think it would benefit from a short note to discuss why there had been a change in the killing assay away from KT2442 to this new bacterial target.

Line 126: Again, in this sentence they use the term "outcompete" when it's clearly was killing the *P. aureofaciens*. I think it would be better to note killing rather than competition in such examples.

Line 210: it is a bit disconcerting to hear that they were not able to restore killing by complementation. I do not recall there being a discussion of this negative finding.

Line 236 through 238. Upon initially reading the sentence, I found it surprising and unexpected, since it did not seem like the simple process of killing a bacterium would cause it to disappear from the biofilm. It was only later when the authors noted that this replacement of the dead cells by the IsoF strain only occurred in flow chambers where turbulence etc could in fact have dislodged the dead cells. That said, after reading the entire manuscript I was left with a feeling that there is a bit of an

4overstatement in the authors conclusions that the IsoF strain could invade biofilms, as this seemed to have been restricted to the flow cells where there is some ability to remove cells from the biofilm, and was not observed in pre-established biofilms that had formed on agar plates etc. This made me wonder whether the invasion of biofilms that might have formed on plants and other habitats where they feel the type IV secretion system could be useful in killing target organisms was as likely to occur as they suggest. I feel a bit of further clarification of the situations in which biofilm invasion/replacement is likely to occur is warranted.

There were several aspects of the plant protection experiment that I found a bit awkward or simplistic. The experiment, using small seedlings growing on agar plates is a somewhat unnatural setting, and one that might have facilitated both microbial survival on, and multiplication on the damaged plant tissue. The experiment was done in a way that would have maximized the potential effect of the presence of IsoF together with the pathogen, since they were mixed together and then immediately applied to the wounded plant. There thus would have been maximal opportunity for IsoF to have been in close proximity to the pathogen and thus for it to have killed it before the pathogen could have multiplied and caused disease. I was disappointed that the authors did not attempt to do a more biologically relevant experiment in which plants grown in some sort of soil matrix or even sand would have had the soil flooded simultaneously with a mixture of the bacteria, or even better, flooded first with the pathogen and then shortly thereafter with the beneficial strain. This would have been a more powerful test of the ability of IsoF to kill developing biofilms on the roots, thereby alleviating the likelihood of disease. These would have been quite simple studies to perform and would better reflect the normal process of infection. I was surprised at the way the authors presented the information on the effects of IsoF on the disease process, relating root weight, leaf area, chlorophyll content etc in a PCO plot in figure 5C. I had never seen such results presented in this way, and found it very difficult to interpret. I feel it would have been much cleaner to have simply prepared a small table showing the effect of the three or four treatments on each of these dependent variables.

Line 372 to 375: this important statement was not given the weight that I feel it deserved, as it does point to perhaps a limited scenario in which type IVB killing will prove to be biologically important.

Line 388 - 389: this sentence seems to be a bit of an overstatement given what they have just noted on lines 372 to 375.

I found the discussion a very interesting and pleasant read, although I had the strong impression that it reiterated almost word for word statements that had been made in the results. Some condensation seems possible.

Reviewer #2 (Remarks to the Author):

The manuscript by Purtschert-Montenegro et al describes the phenomenon of contact-dependent killing of other bacterial species by the soil bacterium *Pseudomonas putida* IsoF, shown to be mediated by a Type IV secretion system coded by the newly named kib cluster. This phenomenon has

5been shown before in *Xanthomonas* and *Stenotrophomonas* bacteria. The difference here is that the *P. putida* isoF T4SS belongs to the larger Type IVB class while that of *X. citri* and *S. maltophilia* belong to the Type IVA systems. The authors use fluorescently-labelled knockout strains to show that the killing is dependent on structural components of the T4SS apparatus as well as on a putative effector coded by the *kib* locus. The authors go on to perform some elegant experiments that show that the *kib* locus, and specifically the effector *Piso_02333*, allow the IsoF strain to invade biofilms formed by the *P. putida* K2442 strain (that does not carry a T4SS, but rather a bacteria-killing T6SS). They then show that the IsoF strain can prevent the phytopathogenic bacterium *Ralstonia solanacearum* from infecting tomato seedlings. The manuscript is very well written, the results are clearly explained and presented and the experiments appear to have been carefully carried out. I do, however have a few comments and suggestions that I believe should be addressed.

Perhaps my major criticism (easily addressed) is that the reader is not provided with the accession numbers of the *P. putida* IsoF genome nor the accession numbers of the genes coded by the *kib* locus. Therefore the reader cannot confer whether the genes described in the paper do in fact correspond to the specified T4SSB components. For example, Lines 139-145: Here the *kib* cluster coding 61 genes is described with reference to Figure 2b and Extended Data Table 1. However, in neither this figure or this table, nor anywhere else in the manuscript or in the Supplementary Information are the accession numbers provided for the *Pseudomonas putida* IsoF strain genome or the genes in the *kib* cluster. I searched for these sequences in the NCBI database and could not find any of them. Also, searching for the gene names *Piso_02333* and *Piso_02332* in these databases did not turn up any hits. Finally, no Data Availability Statement was provided, which should contain these accession codes. Is it possible case that the genome of this organism has not yet been made publicly available? I think that it should definitely be deposited and released before being sent off for review. Especially since this strain has been the subject of study by the Eberle group for at least two decades (for example: Steidle et al, 2001. doi: 10.1128/AEM.67.12.5761-5770.2001.). The authors should provide a table with all accession numbers of the 61 genes in the *kib* locus. If this is not possible, then the full nucleotide and translated protein sequences each open reading frame in the *kib* locus should be provided in fasta format as supplementary information.

Another point, in a way related to that above, is that the nature of the *Piso_02332/Piso_02333* pair is not explored to any significant extent. How many homologs of these two proteins are found in the public databases and what can we say about their phylogenetic distribution (individually or as a pair)? Is the *Piso_02332* immunity protein expected to be localized in the cytosol of the periplasm (does it have a signal peptide?). This could provide clues regarding the site of action of its cognate effector. Regarding the *Piso_02333* effector: Does it have any putative motifs that could be used as a recognition signal for secretion by the T4SSB apparatus?

Other points.

Lines 67-69. A bacterial killing T4SS has been characterized in *X. citri* and *Stenotrophomonas maltophilia* (Souza et al, 2015; Bayer-Santos et al, 2019). And homologous systems have been identified in over a hundred other bacterial species (Sgro et al, 2019).

Lines 71-73: The authors imply that DNA transfer is mediated only by class A systems while class B

6systems have until now been restricted to the role of transferring effectors into eukaryotic cells. This is not quite true. For example, Class B T4SSs are responsible for the horizontal transfer (conjugation) of IncI plasmids.

Lines 113-116: Eight out of sixteen killing defective mutants localize to four genes in the kib cluster. What about the other eight insertion mutants? What were their insertion sites?

Lines 103,106-107 and 296: in several places in the manuscript, the term "host range" is used to represent competitor bacterial species. This is not really appropriate. A more suitable term would be "range of target species" or "range of target organisms".

Line 162: the references 22, 32, 55 and 66 refer to studies on only T6SS effector-immunity protein pairs. However, a few thousand bacteria-killing T4SSA effector-immunity protein pairs have been identified in the genomes of over a hundred species.

Lines 202-204. It is reported that the delta32-33 strain grows more slowly than the wild-type strain or the delta33 strain. This could suggest that Piso_02332 may be neutralizing more than one effector.

Lines 208-211: Why were you not able to restore killing of *P. aureofaciens* and KT2442 by complementing the mutant delta32-33 strain with a plasmid pBBR::32-33 coding for the effector-immunity pair. Did the authors confirm that this plasmid in fact expresses the two proteins?

Lines 226-229: It is written that after 3 days of incubation, isoF had formed a mature biofilm by invading and replacing the KT2442 biofilm. This would correspond to time point 120 hours (48 hrs of KT2442 growth on its own plus 72 after addition of IsoF). No Figure is mentioned to show this. Figures 4a and 4b only show growth up to two days after isoF addition to KT2442 biofilms (total of 96 hours).

Lines 281-283: In Figure 5d, why are the deltaT4B CFUs the same as the IsoF CFUs? I would expect the IsoF numbers to be greater than the deltaT4B numbers in these experiments.

Lines 298-302: This is only partially correct. All of the Xanthomonadaceae-like bacterial killing T4SSs have one or more effector/immunity pairs at the same locus containing all of the structural components of the secretion system PLUS other effector/immunity pairs found in other chromosomal locations (Sgro et al, 2019). Here, since we have no access the genome sequence of the IsoF strain under study, we have no way to investigate whether other possible effector/immunity pairs are found in the genome.

Reviewer #3 (Remarks to the Author):

This study reports that a *P. putida* strain contains a locus encoding a typeIVB secretion system which is involved in bacterial killing via cell-cell contact. This system possesses the novel feature since typeIV SS have been reported thus far to only infect eukaryotic cells. Other aspects of this work include that the locus is most likely found in a genomic island which has been recently acquired and

7that this locus is not widespread since it is only found thus far in a bunch of *Pseudomonas* isolates. This work reports on an interesting novel locus found in a plant associated bacterium with a possible role in competition as well as biocontrol. This work is well performed, described and discussed in the context of microbial ecology. The weakness is the lack of mechanistic data and insight on the mechanism(s) of this system and these initial interesting results raise several questions which need more attention in order to make this work more tangible and less 'preliminary'.

Apparently, the method of killing is not via lysis however no insight on possible targets or mechanism is provided; any data on this aspect would considerably increase the impact of this article. The effector is thought to be the 33 locus/protein, has any further experimentation been performed to confirm unequivocally that this is the effector and on its possible target and mechanism?

It is surprising that mutants in this locus significantly affect bacterial growth since these are thought to be accessory present in a recently acquired genomic island. This aspect is unclear and needs more attention/explanation.

No reference is made on whether this system is also able to infect eukaryotic cells or should it be considered specific for bacteria-bacteria interactions? Has this aspect been tested?

The immunity aspect of this system is not entirely clear since two loci appear to be involved; one major locus *Piso_02332* encodes for an immunity protein, however no mechanistic insight is provided and in addition it is not tested whether this locus alone can provide immunity if transferred to other bacteria.

Author Rebuttal to Initial comments

Reviewer #1 (Remarks to the Author):

This very interesting study has done a good job of describing a novel form of contact-dependent killing of bacteria. Contact-dependent killing schemes have become quite the rage in microbial ecology, and yet most have been associated with variants of type VI secretion systems. This study has shown a very novel method of delivering what appear to be toxic effectors using a variant of the type IVB secretion system. The study is quite solid, and the authors have employed a variety of powerful tools including Tn sequencing and various microscopy techniques to both demonstrate the contact dependence of the killing of various other bacteria by the *Pseudomonas putida* strain, and to identify the genetic loci associated with both the toxic effector as well as the immunity function. It is quite remarkable that this type IVB secretion system seems to be quite promiscuous in its ability to kill other bacteria, as they did not seem to find any taxa that were immune to its effect. They might want to comment further on the relative breadth of this killing compared to many other type VI systems that have been studied. As such, there are indeed important translational aspects of the study, as exemplified here in their demonstration that the biological control of bacterial wilt caused by *Ralstonia solanacearum* could be achieved using this strain capable of contact killing. The manuscript was quite

8well written, and the powerful and logical experimental design leading to the discovery of the system was well considered. I have only a few relatively minor comments about both the way certain findings should be described, a few conclusions that may be a bit overstated, as well as suggestions for how the demonstration of plant disease control could have been better assessed and described.

We are very grateful for the positive evaluation of our study and the helpful comments to improve our manuscript.

Specifics:

Lines 67-69. We are told here that a *Xanthomonas* strain had exhibited a type IV dependent killing of other bacterial strains - while it is only in a couple of sentences later that the authors note that there is more than one kind of type IV secretion system, and that there had been no prior demonstration of the type IVB system being used in antimicrobial activities. However, there was never a mention until much later in the discussion that the *Xanthomonas* example, was in fact, a type IVA secretion system. I feel that this point should have been made earlier, because I was bothered by the concern that this study might not have been as novel as they were suggesting, because they had not noted that this earlier example was not a type IVB system.

The criticism was noticed and we now specify that previous work in *Xanthomonas* identified a T4ASS in contrast to the T4BSS we identified in *P. putida* IsoF.

Line 90 and elsewhere. Here and in many locations throughout the manuscript, the authors have used the term "competition" to describe the interaction of IsoF with various other bacteria. At least in my mind, the term "competition" implies some sort of growth reduction that would be a result of the need to share certain nutrient resources with a neighbor, or perhaps an effect of the chemical environment by a neighbor etc. Given that they are demonstrating here that "competition" is actually the direct killing of their neighbor, I wonder if it would be cleaner to refer to "interaction" or "killing" or some other term that does not seem to have this nutrient competition connotation.

We are thankful for this valuable comment. Accordingly, we have changed the term competition to 'interaction' or 'killing' depending on the context throughout the text. We also changed 'contact-dependent competition (CDC)' to 'contact-dependent killing (CDK)'.

Line 97 misspelling: iodide

Corrected

Line 112. I had to take a step back and try to figure out why the authors had decided to use

Pseudomonas aureofaciens in the study discussed in this section because there had not been a previous mention that this particular species was in fact susceptible to the killing effect of strain IsoF. One had to dig into the results of Figure 1, and specifically recognize that *Pseudomonas aureofaciens* was one of the strains that was in fact susceptible to the killing by IsoF. I think it would benefit from a short note to discuss why there had been a change in the killing assay away from KT2442 to this new bacterial target.

Thank you for making us aware of this inconsistency. We have chosen *P. aureofaciens* in these assays because this strain was more sensitive to killing by IsoF than KT2442. We have added this information in the revised version of the manuscript.

Line 126: Again, in this sentence they use the term “outcompete” when it's clearly was killing the *P. aureofaciens*. I think it would be better to note killing rather than competition in such examples.

Changed to kill as suggested.

Line 210: it is a bit disconcerting to hear that they were not able to restore killing by complementation. I do not recall there being a discussion of this negative finding.

To address the concern of the reviewer, we have analyzed the strains by SDS-PAGE in order to investigate ectopic expression of *PisoF_02332* and *PisoF_02333*. In the complemented strain, we observed a band corresponding to the of the immunity protein but could not detect the effector, indicating that the immunity protein is produced in excess over the toxin. We therefore hypothesize that killing was not restored as a consequence of an unphysiological overexpression of the immunity protein in the complemented strain that effectively neutralized all effector molecules. We have added this information to the Results section and show the SDS-PAGE in the new Fig. 8b of the Extended Data.

Line 236 through 238. Upon initially reading the sentence, I found it surprising and unexpected, since it did not seem like the simple process of killing a bacterium would cause it to disappear from the biofilm. It was only later when the authors noted that this replacement of the dead cells by the IsoF strain only occurred in flow chambers where turbulence etc could in fact have dislodged the dead cells. That said, after reading the entire manuscript I was left with a feeling that there is a bit of an overstatement in the authors conclusions that the IsoF strain could invade biofilms, as this seemed to have been restricted to the flow cells where there is some ability to remove cells from the biofilm, and was not observed in pre-established biofilms that had formed on agar plates etc. This made me wonder whether the invasion of biofilms that might have formed on plants and other habitats where they feel the type IV secretion system could be useful in killing target organisms was

as likely to occur as they suggest. I feel a bit of further clarification of the situations in which biofilm invasion/replacement is likely to occur is warranted.

This is a very valuable and interesting comment. To investigate whether IsoF could also invade a biofilm grown on an agar plate we followed the fate of the strain in CDK assays against KT2442 over 72 hours. While growth of IsoF was restricted to the initial inoculation area after 24 hours, we observed that IsoF invaded the space occupied by the target strain and formed satellite colonies after 72 hours. We hypothesize that killed cells eventually lyse and no longer form a barrier that prevents invasion. Another important factor for invasion competence is that IsoF produces the very powerful biosurfactant putisolvin, which was not only shown to disperse pre-established biofilms but also allows the strain to translocate over semisolid surfaces by means of swarming motility. We have added this information in the Results section, as new Fig. 12 in the Extended Data, and amended the discussion to clarify this issue.

There were several aspects of the plant protection experiment that I found a bit awkward or simplistic. The experiment, using small seedlings growing on agar plates is a somewhat unnatural setting, and one that might have facilitated both microbial survival on, and multiplication on the damaged plant tissue. The experiment was done in a way that would have maximized the potential effect of the presence of IsoF together with the pathogen, since they were mixed together and then immediately applied to the wounded plant. There thus would have been maximal opportunity for IsoF to have been in close proximity to the pathogen and thus for it to have killed it before the pathogen could have multiplied and caused disease. I was disappointed that the authors did not attempt to do a more biologically relevant experiment in which plants grown in some sort of soil matrix or even sand would have had the soil flooded simultaneously with a mixture of the bacteria, or even better, flooded first with the pathogen and then shortly thereafter with the beneficial strain. This would have been a more powerful test of the ability of IsoF to kill developing biofilms on the roots, thereby alleviating the likelihood of disease. These would have been quite simple studies to perform and would better reflect the normal process of infection. I was surprised at the way the authors presented the information on the effects of IsoF on the disease process, relating root weight, leaf area, chlorophyll content etc in a PCO plot in figure 5C. I had never seen such results presented in this way, and found it very difficult to interpret. I feel it would have been much cleaner to have simply prepared a small table showing the effect of the three or four treatments on each of these dependent variables.

We agree that the plant protection experiment is somewhat simplistic and artificial. To follow the advice of the reviewer, we established a soil-based infection model as previously described by Medina and López-Baena, 2018 and used it to investigate the effect of the *kib* cluster on biocontrol activity

11under more natural conditions. In this setup non-sterile soil is drenched simultaneously with a mixture of IsoF or $\Delta T4B$ and *Ralstonia*. The results, which are shown in the modified Fig. 5 as well as in the new Fig. 20 of the Extended Data, demonstrated that IsoF successfully prevents disease and that this effect is dependent on the *kib* gene cluster. The results of these experiments were included in the main text. We feel that the PCA graph is the best way to present our data. However, the criticism was noticed and to satisfy the concerns of the reviewer we also added the independent graphs for each parameter measured in Fig. 18 of the Extended Data.

Line 372 to 375: this important statement was not given the weight that I feel it deserved, as it does point to perhaps a limited scenario in which type IVB killing will prove to be biologically important.

As mentioned before, we have added additional data showing that IsoF in fact is capable of invading a pre-established biofilm on an agar plate, it just requires more time (72 h versus 24 h, which we used in our routine assays). We have included this information in Fig. 4c and amended the discussion accordingly. Moreover, we have rephrased our statement on the importance of putisolvin to replace cells in biofilms, as previous work has demonstrated that this biosurfactant can efficiently remove pre-established biofilms (Kuiper et al., 2004).

Line 388 - 389: this sentence seems to be a bit of an overstatement given what they have just noted on lines 372 to 375.

As mentioned above, IsoF not only invaded a pre-established biofilm in flow-cells but was also capable of invading a pre-established biofilm on an agar plate.

I found the discussion a very interesting and pleasant read, although I had the strong impression that it reiterated almost word for word statements that had been made in the results. Some condensation seems possible.

We have tightened up the text as suggested.

Reviewer #2 (Remarks to the Author):

The manuscript by Purtschert-Montenegro et al describes the phenomenon of contact-dependent killing of other bacterial species by the soil bacterium *Pseudomonas putida* IsoF, shown to be mediated by a Type IV secretion system coded by the newly named *kib* cluster. This phenomenon has been shown

12before in *Xanthomonas* and *Stenotrophomonas* bacteria. The difference here is that the *P. putida* isoF T4SS belongs to the larger Type IVB class while that of *X. citri* and *S. maltophilia* belong to the Type IVA systems. The authors use fluorescently-labelled knockout strains to show that the killing is dependent on structural components of the T4SS apparatus as well as on a putative effector coded by the *kib* locus. The authors go on to perform some elegant experiments that show that the *kib* locus, and specifically the effector *Piso_02333*, allow the IsoF strain to invade biofilms formed by the *P. putida* K2442 strain (that does not carry a T4SS, but rather a bacteria-killing T6SS). They then show that the IsoF strain can prevent the phytopathogenic bacterium *Ralstonia solanacearum* from infecting tomato seedlings. The manuscript is very well written, the results are clearly explained and presented and the experiments appear to have been carefully carried out. I do, however have a few comments and suggestions that I believe should be addressed.

Perhaps my major criticism (easily addressed) is that the reader is not provided with the accession numbers of the *P. putida* IsoF genome nor the accession numbers of the genes coded by the *kib* locus. Therefore the reader cannot confer whether the genes described in the paper do in fact correspond to the specified T4SSB components. For example, Lines 139-145: Here the *kib* cluster coding 61 genes is described with reference to Figure 2b and Extended Data Table 1. However, in neither this figure or this table, nor anywhere else in the manuscript or in the Supplementary Information are the accession numbers provided for the *Pseudomonas putida* IsoF strain genome or the genes in the *kib* cluster. I searched for these sequences in the NCBI database and could not find any of them. Also, searching for the gene names *Piso_02333* and *Piso_02332* in these databases did not turn up any hits. Finally, no Data Availability Statement was provided, which should contain these accession codes. Is it possible case that the genome of this organism has not yet been made publicly available? I think that it should definitely be deposited and released before being sent off for review. Especially since this strain has been the subject of study by the Eberle group for at least two decades (for example: Steidle et al, 2001. doi: 10.1128/AEM.67.12.5761-5770.2001.). The authors should provide a table with all accession numbers of the 61 genes in the *kib* locus. If this is not possible, then the full nucleotide and translated protein sequences each open reading frame in the *kib* locus should be provided in fasta format as supplementary information.

We are very sorry for this mistake. The genome sequence of IsoF is now available at NCBI under the accession number CP072013. We have added this in a Data Availability Statement as requested. In addition, we have added the accession numbers of the 61 genes of the *kib* locus in the new Table 4 in the Extended Dataset.

Another point, in a way related to that above, is that the nature of the *Piso_02332/Piso_02333* pair is not explored to any significant extent.

How many homologs of these two proteins are found in the public databases and what can we say about their phylogenetic distribution (individually or as a pair)?

We are thankful for this comment. We searched the NCBI database for homologs of *PisoF_02332* and *PisoF_02333* and found that the operon structure is fully conserved, supporting the idea that they represent an E-I pair. Moreover, the genes were exclusively found within homologs of the *kib* locus in a few *Pseudomonas* strains, we unable to identify orphan homologs. The comparison of the phylogenetic trees of the *PisoF_02332-33* genes, all orthologs of the *kib* cluster and eight housekeeping genes of the strains carrying the *kib* locus revealed that the tree topology is congruent, suggesting that strains carrying the *kib* cluster form a defined lineage that originated from a common ancestor. We have added this information as a new section in the results and in the new Fig. 9 of the Extended Data.

Is the *Piso_02332* immunity protein expected to be localized in the cytosol of the periplasm (does it have a signal peptide?). This could provide clues regarding the site of action of its cognate effector.

Using LocTREE and PSORTb the subcellular localization of *PisoF_02332* and *PisoF_02333* was predicted to be to be cytoplasmic. Additionally, using SignalP-6.0 web tool we found that neither protein had a signal peptide. We have added this information in the results section.

Regarding the *Piso_02333* effector: Does it have any putative motifs that could be used as a recognition signal for secretion by the T4SSB apparatus?

Although the T4BSS of *Legionella* translocates more than 330 effectors only few recognition sequences have been described. For some effectors the following C terminal motifs have been described: hydrophobic residues (Nagai et al., 2005; Voth et al., 2012), an EExxE domain (Huang et al., 2011) and a FxxxLxxxK domain (Kim et al., 2020). However, none of these motifs could be identified in the C-terminal region of the effector *PisoF_02333*. Interestingly, a FxxxLxxxK domain was found to be present in the C-terminal region of the immunity protein *PisoF_02332*, suggesting that this protein may be transferred together with its cognate effector toxin. Moreover, we noticed that *PisoF_02333* has an unusual glutamine-rich domain in the C-terminal region of the protein (9 Q of 14 aa; Extended Data Fig. 8a) and speculate that these may play a role in effector recognition. Interestingly, a conserved glutamine-rich domain was also identified in the C-terminal regions of VirD4 coupling proteins of phylogenetically diverse T4ASSs (Das, 2020). This region was shown to be required for

recognition of T-strand DNA but not of the second transferred substrate, the single-stranded DNA-binding protein VirE2. This information has been added.

Other points.

Lines 67-69. A bacterial killing T4SS has been characterized in *X. citri* and *Stenotrophomonas maltophilia* (Souza et al, 2015; Bayer-Santos et al, 2019). And homologous systems have been identified in over a hundred other bacterial species (Sgro et al, 2019).

We have added this information and references as requested.

Lines 71-73: The authors imply that DNA transfer is mediated only by class A systems while class B systems have until now been restricted to the role of transferring effectors into eukaryotic cells. This is not quite true. For example, Class B T4SSs are responsible for the horizontal transfer (conjugation) of *Incl* plasmids.

We are grateful for this comment and have re-phrased the sentence to indicate that transferring effectors into eukaryotic cells is not the only function of T4BSS.

Lines 113-116: Eight out of sixteen killing defective mutants localize to four genes in the *kib* cluster. What about the other eight insertion mutants? What were their insertion sites?

We were not able to amplify the regions of the Tn5 insertions in the other 8 mutants and, given that all successfully sequenced mutants were within the *kib* gene cluster, we did not further analyze these mutants.

Lines 103,106-107 and 296: in several places in the manuscript, the term "host range" is used to represent competitor bacterial species. This is not really appropriate. A more suitable term would be "range of target species" or "range of target organisms".

This is a good point and we have changed the wording to 'range of target species' throughout the text.

Line 162: the references 22, 32, 55 and 66 refer to studies on only T6SS effector-immunity protein pairs. However, a few thousand bacteria-killing T4SSA effector-immunity protein pairs have been identified in the genomes of over a hundred species.

This is true, thank you for pointing this out. We have added the following publications in the text: Bayer-Santos et al., 2019; Sgro et al., 2019; Souza et al., 2015.

Lines 202-204. It is reported that the delta32-33 strain grows more slowly than the wild-type strain or the delta33 strain. This could suggest that Piso_02332 may be neutralizing more than one effector.

In fact, we speculated that PisoF_02351 and PisoF_02352 may encode an additional E-I pair. However, recent work in our lab did not support the idea of another E-I pair within the *kib* gene cluster and as a consequence we have removed this statement. Our most recent results rather suggest that the observed retarded growth of the mutant is a consequence of a defective *kib* nanomachinery, which appears to impact viability. In this context, it is interesting to note that inactivation of *dotL* in certain *Legionella pneumophila* strains is lethal (Buscher et al., 2005). However, lethality of a *dotL* mutation is suppressed by mutation of other components of the T4BSS, indicating that the interactions between the different protein components is finely tuned and that a disturbance may lead to self-toxicity.

Lines 208-211: Why were you not able to restore killing of *P. aureofaciens* and KT2442 by complementing the mutant delta32-33 strain with a plasmid pBBR::32-33 coding for the effector-immunity pair. Did the authors confirm that this plasmid in fact expresses the two proteins?

As already mentioned in our response #1, we have analyzed the strains by SDS-PAGE in order to investigate ectopic expression of PisoF_02332 and PisoF_02333. In the complemented strain, we observed a band corresponding to the of the immunity protein but could not detect the toxin, indicating that the immunity protein is produced in excess over the toxin. We therefore hypothesize that killing was not restored because of an unphysiological overexpression of the immunity protein in the complemented strain that effectively neutralized the effector. We have added this information to the discussion and show the SDS-PAGE in the new Fig. 8b of the Extended Data.

Lines 226-229: It is written that after 3 days of incubation, isoF had formed a mature biofilm by invading and replacing the KT2442 biofilm. This would correspond to time point 120 hours (48 hrs of KT2442 growth on its own plus 72 after addition of IsoF). No Figure is mentioned to show this. Figures 4a and 4b only show growth up to two days after isoF addition to KT2442 biofilms (total of 96 hours).

The reviewer is right that the data shown only cover 72 hours of co-cultivation. The text was corrected accordingly. Thank you for pointing this out.

Lines 281-283: In Figure 5d, why are the deltaT4B CFUs the same as the IsoF CFUs? I would expect the IsoF numbers to be greater than the deltaT4B numbers in these experiments.

IsoF was shown to be an excellent tomato root colonizer that forms biofilms on the root surface (Steidle et al., 2001). We speculate that that the constant CFU numbers reflect that the loading capacity of the root for IsoF has been reached and that this is not affected by the *kib* locus. In this context, it is also important to keep in mind that *Ralstonia* only transiently colonizes the root surface before the bacteria gain access to host root systems through natural wounds caused by the emergence of lateral roots or through wounds acquired as roots grow through the soil (Xue et al., 2020).

Lines 298-302: This is only partially correct. All of the Xanthomonadaceae-like bacterial killing T4SSs have one or more effector/immunity pairs at the same locus containing all of the structural components of the secretion system PLUS other effector/immunity pairs found in other chromosomal locations (Sgro et al, 2019). Here, since we have no access the genome sequence of the IsoF strain under study, we have no way to investigate whether other possible effector/immunity pairs are found in the genome.

We are sorry that the genome sequence was not available at the time of submission. We have made it available now. We have reworded the statement and shortened the text.

Reviewer #3 (Remarks to the Author):

This study reports that a *P. putida* strain contains a locus encoding a type IVB secretion system which is involved in bacterial killing via cell-cell contact. This system possesses the novel feature since type IV SS have been reported thus far to only infect eukaryotic cells. Other aspects of this work include that the locus is most likely found in a genomic island which has been recently acquired and that this locus is not widespread since it is only found thus far in a bunch of *Pseudomonas* isolates.

This work reports on an interesting novel locus found in a plant associated bacterium with a possible role in competition as well as biocontrol. This work is well performed, described and discussed in the context of microbial ecology. The weakness is the lack of mechanistic data and insight on the mechanism(s) of this system and these initial interesting results raise several questions which need more attention in order to make this work more tangible and less 'preliminary'.

We are thankful for the supportive comments. We agree with the reviewer that it would be nice to have more insights into the underlying molecular mechanisms of killing by this nanomachinery. The main difficulty in this respect is the absolute novelty of the system that shares no homology with any other killing machinery. However, the criticism was noticed and we have performed additional

17experiments to shed more light on the mode of killing. However, the main focus of our study was indeed the identification and characterization of this novel killing machinery and the evaluation of its biocontrol potential.

Apparently, the method of killing is not via lysis however no insight on possible targets or mechanism is provided; any data on this aspect would considerably increase the impact of this article. The effector is thought to be the 33 locus/protein, has any further experimentation been performed to confirm unequivocally that this is the effector and on its possible target and mechanism?

Bioinformatic analysis of *PisoF_02332* and *PisoF_02333* revealed that both proteins are located in the cytoplasm. We have added this information to the text and in the new Fig. 8 of the Extended Data. As mentioned in our response to reviewer #2, we also noticed that *PisoF_02333* has an unusual glutamine-rich domain in the C-terminal region of the protein (9 Q of 14 aa; Extended Data Fig. 8a) and speculate that it may play a role in effector recognition. Interestingly, a conserved glutamine-rich domain was also identified in the C-terminal regions of *VirD4* coupling proteins of phylogenetically diverse T4ASSs. This region was shown to be required for recognition of T-strand DNA but not of the second transferred substrate, the single-stranded DNA-binding protein *VirE2* (Das, 2020). This information has been added to the Discussion.

Moreover, work that is currently ongoing in the lab aims at isolating resistant mutants of susceptible bacteria with the aim to identify the molecular target(s) of *kib*-encoded effectors. If successful, these investigations may allow us to gain insights into the mode of killing. However, we feel that this will be a story on its own.

It is surprising that mutants in this locus significantly affect bacterial growth since these are thought to be accessory present in a recently acquired genomic island. This aspect is unclear and needs more attention/explanation.

As discussed before, we hypothesize that the observed retarded growth of the mutant is a consequence of a defective *kib* nanomachinery, which may impact viability. It appears that the interactions between the different protein components of *kib* nanomachinery is finely tuned and that a disturbance by deleting the E-I pair may lead to self-toxicity. We have added this information to the Discussion to improve clarity.

No reference is made on whether this system is also able to infect eukaryotic cells or should it be considered specific for bacteria-bacteria interactions? Has this aspect been tested?

We have tested IsoF in a *C. elegans* infection model and found the strain to be avirulent. Likewise, we could not observe antifungal or anti-oomycete activities. It is also noteworthy that IsoF does not grow at 37 °C and thus will be unable to infect mammalian cells. Furthermore, in our tomato root colonization assays we could not observe an effect of the *kib* locus on plant growth. Collectively, these data suggest that this killing machinery is specific for bacterial cells interactions.

The immunity aspect of this system is not entirely clear since two loci appear to be involved; one major locus *Piso_02332* encodes for an immunity protein, however no mechanistic insight is provided and in addition it is not tested whether this locus alone can provide immunity if transferred to other bacteria.

We showed that complementation of the $\Delta 32-33$ mutant by providing *PisoF_02332* *in trans* on a plasmid (pBBR::32) rendered the strain resistant to killing by the wildtype. However, we were unable to complement the $\Delta T4B$ mutant, which lacks the entire *kib* gene cluster, with pBBR::32. Analysis of the strains by SDS-PAGE revealed that a band corresponding to the *PisoF_02332* protein is visible in the complemented $\Delta 32-33$ mutant but not the complemented $\Delta T4B$ mutant. The lack of *PisoF_02332* expression in the $\Delta T4B$ mutant background explains its sensitivity to *kib*-mediated killing and suggests that the *kib* gene cluster encodes functions required for the expression of *PisoF_02332* or affects its stability. We have added this information to the main text and have removed the speculation that another E-I pair is encoded by the cluster, as work currently under way in our laboratory does not support this idea.

Decision Letter, first revision:

Dear Dr. Eberl,

Thank you for submitting your revised manuscript "A type IVB secretion system adapted for bacterial killing, biofilm invasion and biocontrol" (NMICROBIOL-21092434A). It has now been seen by the original referees and their comments are below. The reviewers find that the paper has improved in revision, and therefore we'll be happy in principle to publish it in Nature Microbiology, pending minor revisions to satisfy the referees' final requests and to comply with our editorial and formatting guidelines.

19Thank you again for your interest in Nature Microbiology Please do not hesitate to contact me if you have any questions.

Sincerely,

{redacted}

--

Reviewer #1 (Remarks to the Author):

see below

Reviewer #3 (Remarks to the Author):

This revised version is an improvement as authors have carefully revised most of the points raised by the three reviewers.

Most additional experimenting and data was also provided:

- SDS PAGE analysis, the immunity protein was believed to be under-expressed compared to the toxin hence there was a lack of complementation.
- The authors have performed a new plant experiment based on a soil infection model in order to test biocontrol activity under conditions which resemble more the wild. The results confirmed what was previously observed in more controlled conditions.
- An additional biofilm experiment was also performed involving a longer time frame further evidencing the effects of bacterial invasion.
- The genome sequence was properly deposited in data banks.
- Performed a more complete phylogenetic search of the two ORFs (2332 and 2333)
- More in silico protein information of the 2332 and 2333 ORFs
- Have tested the killing model on *C. elegans* indicating that it does not appear to infect eukaryotic cells

In the revised text, these new additional results and figures are well integrated in the document and have no comments.

Minor comment:

Not clear the reasoning why 8 Tn5 mutants could not be mapped – nowadays with NGS it is ultimately possible to map any mutation in the genome. I suggest to either map these mutants or just remove this information and simply state that a subset was mapped.

Decision Letter, final checks

Dear Dr. Eberl,

Thank you for your patience as we've prepared the guidelines for final submission of your Nature Microbiology manuscript, "A type IVB secretion system adapted for bacterial killing, biofilm invasion and biocontrol" (NMICROBIOL-21092434A). Please carefully follow the step-by-step instructions provided in the attached file, and add a response in each row of the table to indicate the changes that you have made. Please also check and comment on any additional marked-up edits we have proposed within the text. Ensuring that each point is addressed will help to ensure that your revised manuscript can be swiftly handed over to our production team.

In recognition of the time and expertise our reviewers provide to Nature Microbiology's editorial process, we would like to formally acknowledge their contribution to the external peer review of your manuscript entitled "A type IVB secretion system adapted for bacterial killing, biofilm invasion and biocontrol". For those reviewers who give their assent, we will be publishing their names alongside the published article.

Nature Microbiology offers a Transparent Peer Review option for new original research manuscripts submitted after December 1st, 2019. As part of this initiative, we encourage our authors to support increased transparency into the peer review process by agreeing to have the reviewer comments, author rebuttal letters, and editorial decision letters published as a Supplementary item. When you submit your final files please clearly state in your cover letter whether or not you would like to participate in this initiative. Please note that failure to state your preference will result in delays in accepting your manuscript for publication.

Cover suggestions

As you prepare your final files we encourage you to consider whether you have any images or illustrations that may be appropriate for use on the cover of Nature Microbiology.

21Covers should be both aesthetically appealing and scientifically relevant, and should be supplied at the best quality available. Due to the prominence of these images, we do not generally select images featuring faces, children, text, graphs, schematic drawings, or collages on our covers.

Nature Microbiology has now transitioned to a unified Rights Collection system which will allow our Author Services team to quickly and easily collect the rights and permissions required to publish your work. Approximately 10 days after your paper is formally accepted, you will receive an email in providing you with a link to complete the grant of rights. If your paper is eligible for Open Access, our Author Services team will also be in touch regarding any additional information that may be required to arrange payment for your article.

Please note that *Nature Microbiology* is a Transformative Journal (TJ). Authors may publish their research with us through the traditional subscription access route or make their paper immediately open access through payment of an article-processing charge (APC). Authors will not be required to make a final decision about access to their article until it has been accepted. [Find out more about Transformative Journals](https://www.springernature.com/gp/open-research/transformative-journals)

Authors may need to take specific actions to achieve [compliance with funder and institutional open access mandates](https://www.springernature.com/gp/open-research/funding/policy-compliance-faqs). If your research is supported by a funder that requires immediate open access (e.g. according to [Plan S principles](https://www.springernature.com/gp/open-research/plan-s-compliance)) then you should select the gold OA route, and we will direct you to the compliant route where possible. For authors selecting the subscription publication route, the journal's standard licensing terms will need to be accepted, including [self-archiving policies](https://www.nature.com/nature-portfolio/editorial-policies/self-archiving-and-license-to-publish). Those licensing terms will supersede any other terms that the author or any third party may assert apply to any version of the manuscript.

For information regarding our different publishing models please see our page

href="https://www.springernature.com/gp/open-research/transformational-journals"> Transformational Journals page. If you have any questions about costs, Open Access requirements, or our legal forms, please contact ASJournals@springernature.com.

Please use the following link for uploading these materials:
{redacted}

Best regards,

{redacted}

Reviewer #1:
Remarks to the Author:
see below

Reviewer #3:
Remarks to the Author:

This revised version is an improvement as authors have carefully revised most of the points raised by the three reviewers.

Most additional experimenting and data was also provided:

- SDS PAGE analysis, the immunity protein was believed to be under-expressed compared to the toxin hence there was a lack of complementation.
- The authors have performed a new plant experiment based on a soil infection model in order to test biocontrol activity under conditions which resemble more the wild. The results confirmed what was previously observed in more controlled conditions.
- An additional biofilm experiment was also performed involving a longer time frame further evidencing the effects of bacterial invasion.
- The genome sequence was properly deposited in data banks.
- Performed a more complete phylogenetic search of the two ORFs (2332 and 2333)
- More in silico protein information of the 2332 and 2333 ORFs
- Have tested the killing model on *C. elegans* indicating that it does not appear to infect eukaryotic cells

In the revised text, these new additional results and figures are well integrated in the document and have no comments.

Minor comment:

Not clear the reasoning why 8 Tn5 mutants could not be mapped – nowadays with NGS it is ultimately possible to map any mutation in the genome. I suggest to either map these mutants or just remove this information and simply state that a subset was mapped.

Final Decision Letter:

Dear Leo,

I am pleased to accept your Article "*Pseudomonas putida* mediates bacterial killing, biofilm invasion and biocontrol with a type IVB secretion system" for publication in Nature Microbiology. Thank you for having chosen to submit your work to us and many congratulations.

Acceptance of your manuscript is conditional on all authors' agreement with our publication policies (see <https://www.nature.com/nmicrobiol/editorial-policies>). In particular your manuscript must not be published elsewhere and there must be no announcement of the work to any media outlet until the publication date (the day on which it is uploaded onto our website).

Please note that *Nature Microbiology* is a Transformative Journal (TJ). Authors may publish their research with us through the traditional subscription access route or make their paper immediately open access through payment of an article-processing charge (APC). Authors will not be required to make a final decision about access to their article until it has been accepted. [Find out more about Transformative Journals](https://www.springernature.com/gp/open-research/transformative-journals)

24Authors may need to take specific actions to achieve [compliance](https://www.springernature.com/gp/open-research/funding/policy-compliance-faqs) with funder and institutional open access mandates. If your research is supported by a funder that requires immediate open access (e.g. according to [Plan S principles](https://www.springernature.com/gp/open-research/plan-s-compliance)) then you should select the gold OA route, and we will direct you to the compliant route where possible. For authors selecting the subscription publication route, the journal's standard licensing terms will need to be accepted, including [self-archiving policies](https://www.nature.com/nature-portfolio/editorial-policies/self-archiving-and-license-to-publish). Those licensing terms will supersede any other terms that the author or any third party may assert apply to any version of the manuscript.
